# Why should autoencoders work?

**Matthew D. Kvalheim**                                    *kvalheim@umbc.edu*
*Department of Mathematics and Statistics*
*University of Maryland, Baltimore County, MD, United States.*

**Eduardo D. Sontag**                                      *sontag@sontaglab.org*
*Departments of Electrical and Computer Engineering and Bioengineering*
*Northeastern University, Boston, MA, United States.*

Reviewed on OpenReview: *https://openreview.net/forum?id=uGVFtjvI3v*

## Abstract

Deep neural network autoencoders are routinely used computationally for model reduction. They allow recognizing the intrinsic dimension of data that lie in a $k$-dimensional subset $K$ of an input Euclidean space $\mathbb{R}^n$. The underlying idea is to obtain both an encoding layer that maps $\mathbb{R}^n$ into $\mathbb{R}^k$ (called the bottleneck layer or the space of latent variables) and a decoding layer that maps $\mathbb{R}^k$ back into $\mathbb{R}^n$, in such a way that the input data from the set $K$ is recovered when composing the two maps. This is achieved by adjusting parameters (weights) in the network to minimize the discrepancy between the input and the reconstructed output. Since neural networks (with continuous activation functions) compute continuous maps, the existence of a network that achieves perfect reconstruction would imply that $K$ is homeomorphic to a $k$-dimensional subset of $\mathbb{R}^k$, so clearly there are topological obstructions to finding such a network. On the other hand, in practice the technique is found to "work" well, which leads one to ask if there is a way to explain this effectiveness. We show that, up to small errors, indeed the method is guaranteed to work. This is done by appealing to certain facts from differential topology. A computational example is also included to illustrate the ideas.

## 1 Introduction

Many real-world problems require the analysis of large numbers of data points inhabiting some Euclidean space $\mathbb{R}^n$. The "manifold hypothesis" (Fefferman et al., 2016) postulates that these points lie on some $k$-dimensional submanifold with (or without) boundary $K \subseteq \mathbb{R}^n$, so can be described locally by $k < n$ parameters. When $K$ is a linear submanifold, classical approaches like principal component analysis and multidimensional scaling are effective ways to learn these parameters. But when $K$ is nonlinear, learning these parameters is the more challenging "manifold learning" problem studied in the rapidly developing literature on "geometric deep learning" (Bronstein et al., 2017).

One popular approach to this problem relies on deep neural network **autoencoders** (also called "replicators" (Hecht-Nielsen, 1995)) of the form $G \circ F$, where the output of the **encoder** $F \colon \mathbb{R}^n \to \mathbb{R}^k$ is the desired $k < n$ parameters, $G \colon \mathbb{R}^k \to \mathbb{R}^n$ is the **decoder**, and $F$ and $G$ are continuous. See Figure 1 for an illustration. The goal is to learn $F, G$ to create a perfect autoencoder, one such that $G(F(x)) = x$ for all $x \in K$. The latter condition implies that $F|_K : K \to F(K) \subseteq \mathbb{R}^k$ is a homeomorphism, since it is a continuous map with a continuous inverse $G \colon F(K) \to K$. Thus, a perfect autoencoder $F, G$ exists if and only if the $k$-dimensional $K$ is homeomorphic to a subset of $\mathbb{R}^k$, so there are topological obstructions making this goal impossible in general, as observed in Batson et al. (2021).

And yet, the wide practical applicability of the method evidences remarkable empirical success from autoencoders even when $K$ is not homeomorphic to such a subset of $\mathbb{R}^k$. (We give an illustrative numerical experiment in §3.) How can this be?

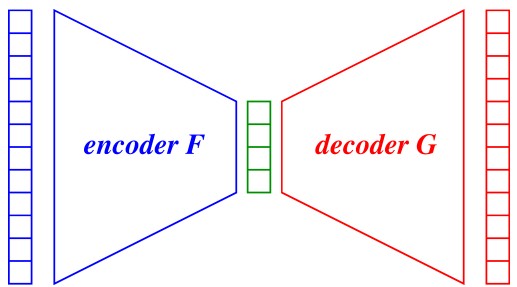

Figure 1: An autoencoder consists of an encoding layer, which maps inputs that lie in a subset $K$ of $\mathbb{R}^n$ ($n = 12$ in this illustration) into a hidden or latent layer of points in $\mathbb{R}^k$ (here $k = 4$), followed by a decoding layer mapping $\mathbb{R}^k$ back into $\mathbb{R}^n$. The goal is to make the decoded vectors (in red) match the data vectors (in blue). In a perfect autoencoder, $G(F(x)) = x$ for all $x$ in $K$. Due to topological obstructions, a more realistic goal is to achieve $G(F(x)) \approx x$ for all $x$ in a large subset of $K$.

This apparent paradox is resolved by the following Theorem 1, which asserts that the set of $x \in K$ for which $G(F(x)) \not\approx x$ can be made arbitrarily small with respect to the "intrinsic measures" $\partial\mu$ and $\mu$ (defined in §B.3) on $\partial K$ and $K$ generalizing length and surface area. For the statement, $\mathcal{F}^{\ell,m}$ denotes any set of continuous functions $\mathbb{R}^\ell \to \mathbb{R}^m$ with the "universal approximation" property that any continuous function $H\colon \mathbb{R}^\ell \to \mathbb{R}^m$ can be uniformly approximated arbitrarily closely on any compact set $L \subseteq \mathbb{R}^\ell$ by some $\tilde{H} \in \mathcal{F}^{\ell,m}$.

**Theorem 1.** Let $k, n \in \mathbb{N}$ and $K \subseteq \mathbb{R}^n$ be a union of finitely many disjoint compact smoothly embedded submanifolds with boundary each having dimension less than or equal to $k$. For each $\delta > 0$ and finite set $S \subseteq K$, there is a closed set $K_0 \subseteq K$ disjoint from $S$ with intrinsic measures $\mu(K_0) < \delta$, $\partial\mu(K_0 \cap \partial K) < \delta$ such that $M \setminus K_0$ is connected for each component $M$ of $K$, and the following property holds. For each $\varepsilon > 0$ there are functions $F \in \mathcal{F}^{n,k}$, $G \in \mathcal{F}^{k,n}$ such that

$$\sup_{x \in K \setminus K_0} \|G(F(x)) - x\| < \varepsilon. \tag{1}$$

In this paper, we adopt the standard convention that manifolds are the special case of manifolds with boundary for which the boundary is empty.

Theorem 1 may be interpreted as a "probably approximately correct (PAC)" theorem for autoencoders complementary to recent PAC theorems obtained in the manifold learning literature (Fefferman et al., 2016; 2018; 2023). Our theorem asserts that, for any finite training set $S$ of data points in $K$, there is an autoencoder $G \circ F$ with error smaller than $\varepsilon$ on $S$ such that the "generalization error" will also be uniformly smaller than $\varepsilon$ on any test data in $K \setminus K_0$.

**Remark 1.** In particular, Theorem 1 applies when $\mathcal{F}^{\ell,m}$ is a collection of possible functions $\mathbb{R}^\ell \to \mathbb{R}^m$ that can be produced by neural networks. Neural networks, particularly in the context of deep learning, have been extensively studied for their ability to approximate continuous functions. Specifically, the Universal Approximation Theorem states that feedforward networks (even with just one hidden layer) can approximate scalar continuous functions on compact subsets of $\mathbb{R}^\ell$ (and thus, componentwise, can approximate vector functions as well), under mild assumptions on the activation function. This result was proved for sigmoidal activation functions in Cybenko (1989) and generalized in Hornik et al. (1989). Upper bounds on the numbers of units required (in single-hidden layer architectures) were given independently in Jones (1992) and Barron (1993) for approximating functions whose Fourier transforms satisfy a certain integrability condition, providing a least-squares error rate $O(n^{-1/2})$, where $n$ is the number of neurons in the hidden layer, and similar results were provided in Donahue et al. (1997) for (more robust to outliers) approximations in $L^p$ spaces with $1 < p < \infty$. Although these theorems show that single-hidden layer networks are sufficient for universal approximation of continuous functions, it is known from practical experience that deeper architectures are often necessary or at least more efficient. There are theoretical results justifying the advantages of deeper networks. For example, Sontag (1992) showed that the approximation of feedback controllers

for non-holonomic control systems and more generally for inverse problems requires more than one hidden layer, and deeper networks (those with more layers) can represent certain functions more efficiently than shallow networks, in the sense that they require exponentially fewer parameters to achieve a given level of approximation (Eldan & Shamir, 2016; Telgarsky, 2016).

**Remark 2.** While the intrinsic measures $\mu$, $\partial\mu$ are a convenient choice for the statement of Theorem 1, Theorem 1 still holds verbatim if $\mu$, $\partial\mu$ are replaced by any finite Borel measures $\nu$, $\partial\nu$ that are absolutely continuous with respect to $\mu$, $\partial\mu$, respectively. Moreover, Dr. Joshua Batson suggested to us the observation that Theorem 1 implies that the $L^2(\nu)$ loss

$$\int_K \|G(F(x)) - x\|^2 \, d\nu(x)$$

can always be made arbitrarily small (this includes the case $\nu = \mu$). See Remarks 7, 8 in §2 for a detailed explanation of these observations and their implications for autoencoder training.

**Remark 3.** The fact that one can pick $M \setminus K_0$ to be connected for each component $M$ of $K$, which implies that also each encoded "good set" $F(M \setminus K_0)$ is connected, makes Theorem 1 particularly informative and interesting. For example, suppose that our data manifold $K$ is connected. Then Theorem 1 guarantees that $K \setminus K_0$ is connected. In particular, this property implies the ability to "walk along" $K \setminus K_0$ (e.g., for interpolation of images represented by points in $K$) using the latent space, since for each $x, y \in K \setminus K_0$ it implies the existence of a smooth path $t \mapsto \gamma(t)$ from $F(x)$ to $F(y)$ in $F(K \setminus K_0)$ such that, up to $\varepsilon$-small errors, the decoded path $G(\gamma(t))$ goes from $x$ to $y$ while staying in $K$. Also, if this property were not claimed, then a much simpler proof could be based on splitting up $K$ (up to a set of measure zero) into a potentially large number of submanifolds and patching together autoencoders for each piece.

**Remark 4.** One should emphasize that Theorem 1 is a statement about the fundamental capabilities of autoencoders, but it does not imply that numerical learning algorithms will always succeed at finding an autoencoder that satisfies the connectedness constraint (or the desired bounds, for that matter). Our numerical experiments illustrate this phenomenon. For example, Figure 5 shows a learning run in which the encoded good set is (up to sampling resolution) connected, but Figure 8 shows a learning instance in which it is not.

The remainder of the paper is organized as follows. Theorem 1 is proved in §2. The numerical experiments are in §3. A result ruling out certain extensions of Theorem 1 is proved in §4. §5 includes further discussion and directions for future work. An appendix contains the implementation code for these experiments. Another appendix reviews some notions of topology and related concepts that are used in the paper.

## 2 Proof of Theorem 1

In this section we prove Theorem 1. See Appendix B (§B.3) for a description of the notions of "intrinsic measure" and "measure zero" discussed herein. An outline of our strategy for the proof is as follows.

First (Lemma 1), when $K$ consists of a single $k$-dimensional component, we construct a subset $C \subseteq K$ such that $C$ is closed and has measure zero in $K$, $C \cap \partial K$ has measure zero in $\partial K$, and $K \setminus C$ is connected and admits a smooth embedding $K \setminus C \hookrightarrow \mathbb{R}^k$. We next show (Lemma 2) that $C$ can additionally be chosen disjoint from any given finite subset $S \subseteq K$. Successive application of these results extend them to the case that $K$ consists of at most finitely many components, each having dimension less than or equal to $k$. We then construct (Lemma 3) a suitable "thickening" $K_0 \subseteq K$ of $C$ that has arbitrarily small positive intrinsic measure, but otherwise satisfies the same properties as $C$. This thickening is such that the restriction of the smooth embedding $K \setminus C \hookrightarrow \mathbb{R}^k$ to $K \setminus K_0$ extends to a smooth "encoder" map $\tilde{F} \colon \mathbb{R}^n \to \mathbb{R}^k$. Defining the smooth "decoder" map $\tilde{G} \colon \mathbb{R}^k \to \mathbb{R}^n$ to be any smooth extension of the inverse $(\tilde{F}|_K)^{-1} \colon F(K) \to K \subseteq \mathbb{R}^n$ yields an autoencoder with perfect reconstruction on $K \setminus K_0$, i.e., $\tilde{G}(\tilde{F}(x)) = x$ for all $x \in K_0$. Finally, since we consider neural network (or other) function approximators that can uniformly approximate—but not exactly reproduce—all such functions on compact sets, we prove (Theorem 1) that sufficiently close approximations $F$, $G$ of $\tilde{F}$, $\tilde{G}$ will make $\|G(F(x)) - x\|$ arbitrarily uniformly small for all $x \in K \setminus K_0$.

The proof of Lemma 1 constructs $C$ as a union of "stable manifolds" $W^s(q)$ of equilibria $q$ of a certain gradient vector field (§B.3). Such stable manifolds are fundamental in Morse theory (Pajitnov, 2006).

**Lemma 1.** Let $M$ be a $k$-dimensional connected compact smooth manifold with boundary. There exists a set $C \subseteq M$ such that $C$ is closed and has measure zero in $M$, $C \cap \partial M$ has measure zero in $\partial M$, and $M \setminus C$ is connected and admits a smooth embedding into $\mathbb{R}^k$.

**Remark 5.** If $\partial M = \varnothing$, an alternative proof equips $M$ with any Riemannian metric, fixes $p \in M$, and defines $C \subseteq M$ to be the cut locus (Sakai, 1996, Def. III.4.3) with respect to $p$ and the metric. This $C$ is closed and has measure zero, and $M \setminus C$ is diffeomorphic to $\mathbb{R}^k$ (Sakai, 1996, Lem. III.4.4).

*Proof.* **Step 1 (setup).** Fix any $p \in \text{int}(M)$ and Riemannian metric on $M$. There is a smooth function $\varphi \colon M \to [0,1]$ such that all equilibria of the negative gradient vector field $-\nabla\varphi$ are hyperbolic (§B.3) and belong to $\text{int}(M)$, $\{p\} = \varphi^{-1}(0)$ is the unique local minimum, and $\partial M = \varphi^{-1}(1)$ (Koditschek & Rimon, 1990, Thm 3).

**Step 2 (construction of $C$).** Define

$$C := \bigcup \{W^s(q) \colon \nabla\varphi(q) = 0, q \neq p\},$$

where $W^s(q) \subseteq M$ is the set of points whose $(-\nabla\varphi)$-trajectories converge to the equilibrium $q \in \text{int}(M)$ as $t \to \infty$.

**Step 3 ($C$ is closed and connected).** The complement $W^s(p) = M \setminus C$ of $C$ is open and connected since it is the basin of attraction of $p$ for $-\nabla\varphi$ (§B.3), so $C$ is closed.

**Step 4 ($C$ has measure zero).** Since $C \subseteq M$ is a union of smoothly embedded submanifolds with boundary $N$ satisfying $\dim N < \dim M$ and $\partial N = N \cap \partial M$ (Pajitnov, 2006, Prop. 1.3.2.13), $C$ and $C \cap \partial M$ have measure zero in $M$ and $\partial M$, respectively.

**Step 5 ($C$ admits a smooth embedding into $\mathbb{R}^k$).** Finally, since $\text{int}(M) \setminus C$ is the basin of attraction of $p$ for the rescaled vector field $-(1-\varphi)(\nabla\varphi)$ there is a diffeomorphism $F \colon \text{int}(M) \setminus C \approx \mathbb{R}^k$ (Wilson, 1967, Thm 3.4), so if $\Phi^1 \colon M \to \text{int}(M)$ is the smooth embedding sending points $x(0) \in M$ to the values $x(1)$ of their $(-\nabla\varphi)$-trajectories $x(t)$, then $F \circ \Phi^1 \colon M \setminus C \hookrightarrow \mathbb{R}^k$ is the desired smooth embedding. $\qquad\square$

The proof of Lemma 2 constructs a "diffeotopy", a smooth 1-parameter family of diffeomorphisms, that moves $C$ to a subset disjoint from $S$ satisfying the same properties as $C$. The use of diffeotopies (or "ambient isotopies") is a standard technique in differential topology (Hirsch, 1994, Ch. 8).

**Lemma 2.** In the setting of Lemma 1, $C$ can be chosen disjoint from any finite subset $S \subseteq M$.

*Proof.* If $M$ is diffeomorphic to a point or an interval, then $C$ can be taken to be the empty set. If $M$ is diffeomorphic to a circle, then $C$ can be taken to be any point disjoint from $S$. It remains only to consider the case that $\dim M \geq 2$ (Lee, 2013, Ex. 15-13). Since Lemma 1 implies that $C$ does not contain any component of $\partial M$, there is a diffeotopy $\partial J_t$ of $\partial M$, $t \in [0,1]$, such that the image of $S \cap \partial M$ under the diffeomorphism $\partial J_1 \colon \partial M \to \partial M$ does not intersect $C$, that is, it satisfies $\partial J_1(S \cap \partial M) \cap C = \varnothing$ (Hirsch, 1994, p. 186), (Michor & Vizman, 1994). The diffeotopy $\partial J_t$ extends to one generating a diffeotopy $J_t$ of $M$, $t \in [0,1]$, such that the diffeomorphism $J_1 \colon M \to N$ satisfies $J_1(S) \cap C = \varnothing$ (Michor & Vizman, 1994), (Hirsch, 1994, Thm 8.1.3, Thm 8.1.4).[1] Hence the image $\tilde{C} := J_1^{-1}(C)$ of $C$ under the diffeomorphism $J_1^{-1}$ is a closed measure zero set disjoint from $S$, $M \setminus \tilde{C}$ is connected, and $C \cap \partial M$ has measure zero in $\partial M$. Moreover, if $F \colon M \setminus C \to N \subseteq \mathbb{R}^k$ is the smooth embedding from the statement of Lemma 1, then $F \circ J_1 \colon M \setminus \tilde{C} \to N \subseteq \mathbb{R}^k$ is a smooth embedding. Upon replacing $C$ with $\tilde{C}$, this finishes the proof. $\qquad\square$

Lemma 3 makes use of the "intrinsic measure" $\mu$ (§B.3) on any union $K$ of smoothly embedded submanifolds of a Euclidean space that is induced by the Riemannian density (Lee, 2013, p. 428) of the restriction of the Euclidean metric to each component of $K$. We use the notation $\partial\mu$ for the intrinsic measure of $\partial K$. Any measure zero subset $C$ of $K$ in the sense of Lee (2013, p. 128) has intrinsic measure $\mu(C) = 0$, and similarly $\partial\mu(C \cap \partial K) = 0$ when $C \cap \partial K$ has measure zero in $\partial K$.

---

[1] If $\partial M = \varnothing$, then $\partial J_t$ is the empty diffeotopy, so any diffeotopy $J_t$ is automatically an extension of $\partial J_t$. Less pedantically, in the case that $\partial M = \varnothing$, there are simply fewer constraints on $J_t$.

**Remark 6.** If $A$ is a measurable subset (§B.2) of an $\ell$-dimensional component $M$ of $K$, then $\mu(A)$ is simply the $\ell$-dimensional volume of $A$. For example, $\mu(A)$ is the length of $A$ when $k = 1$, the surface area of $A$ when $k = 2$, the volume of $A$ when $k = 3$, and so on.

The proof of Lemma 3 follows the outline at the beginning of this section. To ensure that the complements $M \setminus K_0$ of the "thickening" $K_0$ of $C$ within each component $M$ of $K$ are connected, we construct each $M \setminus K_0$ as a connected component of a sufficiently big sublevel set of a suitable function $h \colon K \setminus C \to [0, \infty)$. See Appendix B (§B.3) for a discussion of the smooth extension lemma used in the proof of Lemma 3.

**Lemma 3.** Let $k, n \in \mathbb{N}$ and $K \subseteq \mathbb{R}^n$ be a union of finitely many disjoint compact smoothly embedded submanifolds with boundary each having dimension less than or equal to $k$. For each $\delta > 0$ and finite set $S \subseteq K$, there are smooth functions $F \colon \mathbb{R}^n \to \mathbb{R}^k$, $G \colon \mathbb{R}^k \to \mathbb{R}^n$ and a closed set $K_0 \subseteq K$ disjoint from $S$ such that $\mu(K_0) < \delta$, $\partial\mu(K_0 \cap \partial K) < \delta$, $M \setminus K_0$ is connected for each component $M$ of $K$, and

$$G \circ F|_{K \setminus K_0} = \mathrm{id}_{K \setminus K_0}.$$

*Proof.* Each component $M$ of $K$ is a connected compact smooth manifold with boundary of dimension less than or equal to $k$. Applying Lemmas 1, 2 to each such component yields the existence of a closed set $C \subseteq K$ disjoint from $S$ such that $C$ has measure zero in $K$, $C \cap \partial K$ has measure zero in $\partial K$, and $M \setminus C$ is connected and admits a smooth embedding into $\mathbb{R}^k$ for each component $M$ of $K$. Compressing the images of these smooth embeddings into arbitrarily small disjoint disks by post-composing each with a suitable diffeomorphism of $\mathbb{R}^k$ produces a smooth embedding $F_0 \colon K \setminus C \to \mathbb{R}^k$.

Let $h \colon K \setminus C \to [0, \infty)$ be any continuous function such that $\{h \leq r\}$ is compact for every $r \geq 0$ (Lee, 2013, Prop. 2.28). Arbitrarily select one point in each component of $K$, and let $U_j \subseteq K$ be the open set equal to the union of the components of $\{h < j\}$ containing each of these points. The properties of $h$ imply that the increasing union $\bigcup_{j \in \mathbb{N}} U_j = K \setminus C$. Thus, finiteness of $S$, compactness of $K$, and outer regularity of the intrinsic measures (§B) imply the existence of $N \in \mathbb{N}$ such that $K_0 := K \setminus U_N$ satisfies $K_0 \cap S = \varnothing$, $K_0 \supseteq C$, $\mu(K_0) < \delta$ and $\partial\mu(K_0 \cap \partial K) < \delta$.

Defining $F \colon \mathbb{R}^n \to \mathbb{R}^k$ and $G \colon \mathbb{R}^k \to \mathbb{R}^n$ respectively to be any smooth extensions (Lee, 2013, Lem. 2.26) of $F_0|_{\mathrm{cl}(U_N)}$ and $(F_0|_{\mathrm{cl}(U_N)})^{-1} \colon F_0(\mathrm{cl}(U_N)) \to \mathrm{cl}(U_N) \subseteq \mathbb{R}^n$ completes the proof. $\qquad\square$

Assume given for each $\ell, m \in \mathbb{N}$ a collection $\mathcal{F}^{\ell,m}$ of continuous functions $\mathbb{R}^\ell \to \mathbb{R}^m$ with the following "universal approximation" property: for any $\varepsilon > 0$, compact subset $L \subseteq \mathbb{R}^\ell$, and continuous function $H \colon \mathbb{R}^\ell \to \mathbb{R}^m$, there is $\tilde{H} \in \mathcal{F}^{\ell,m}$ such that $\max_{x \in L} \|H(x) - \tilde{H}(x)\| < \varepsilon$. Equivalently, $\mathcal{F}^{\ell,m}$ is any collection of continuous functions $\mathbb{R}^\ell \to \mathbb{R}^m$ that is dense in the space of continuous functions $\mathbb{R}^\ell \to \mathbb{R}^m$ with the compact-open topology (Hirsch, 1994, Sec. 2.4) discussed in Appendix B (§B.3). We now restate and prove Theorem 1.

**Theorem 1.** Let $k, n \in \mathbb{N}$ and $K \subseteq \mathbb{R}^n$ be a union of finitely many disjoint compact smoothly embedded submanifolds with boundary each having dimension less than or equal to $k$. For each $\delta > 0$ and finite set $S \subseteq K$, there is a closed set $K_0 \subseteq K$ disjoint from $S$ with intrinsic measures $\mu(K_0) < \delta$, $\partial\mu(K_0 \cap \partial K) < \delta$ such that $M \setminus K_0$ is connected for each component $M$ of $K$, and the following property holds. For each $\varepsilon > 0$ there are functions $F \in \mathcal{F}^{n,k}$, $G \in \mathcal{F}^{k,n}$ such that

$$\sup_{x \in K \setminus K_0} \|G(F(x)) - x\| < \varepsilon. \tag{1}$$

*Proof.* Fix a finite set $S \subseteq K$ and $\delta > 0$. Lemma 3 implies the existence of smooth functions $\tilde{F} \colon \mathbb{R}^n \to \mathbb{R}^k$, $\tilde{G} \colon \mathbb{R}^k \to \mathbb{R}^n$ and a closed set $K_0 \subseteq K$ disjoint from $S$ such that $\mu(K_0) < \delta$, $\partial\mu(K_0 \cap \partial K) < \delta$, $M \setminus K_0$ is connected for each component $M$ of $K$, and $\tilde{G} \circ \tilde{F}|_{K \setminus K_0} = \mathrm{id}_{K \setminus K_0}$.

Fix $\varepsilon > 0$. Since $K$ is compact, and by the density of $\mathcal{F}^{n,k}$, $\mathcal{F}^{k,n}$ and continuity of the composition map $(G, F) \mapsto G \circ F$ in the compact-open topologies (Hirsch, 1994, p. 64, Ex. 10(a)), there exist $F \in \mathcal{F}^{n,k}$, $G \in \mathcal{F}^{k,n}$ such that $G \circ F$ is uniformly $\varepsilon$-close to $\tilde{G} \circ \tilde{F}$ on $K$. Since $\tilde{G}(\tilde{F}(x)) = x$ for all $x \in K \setminus K_0$, the functions $F, G$ satisfy (1). This completes the proof. $\qquad\square$

**Remark 7.** The intrinsic measures $\mu$, $\partial\mu$ are a convenient choice for the statement of Theorem 1, but Theorem 1 still holds verbatim if $\mu$, $\partial\mu$ are replaced by any finite Borel measures $\nu$, $\partial\nu$ that are absolutely continuous with respect to $\mu$, $\partial\mu$, respectively. This is because such measures have the property that for each $\delta_1 > 0$ there is $\delta_2 > 0$ such that $\nu(A), \partial\nu(B) < \delta_1$ whenever $\mu(A), \partial\mu(B) < \delta_2$ (Folland, 1999, Thm 3.5).

**Remark 8.** Many practical algorithms for autoencoders, such as the one used to compute the example in §3, attempt to minimize a least-squares loss, in contrast to the supremum norm loss that Theorem 1 guarantees. In a private communication, Dr. Joshua Batson pointed out to us that, as a corollary of Theorem 1, one can also guarantee a global $L^2$ loss. We next develop the argument sketched by Dr. Batson.

Theorem 1 implies that, for any finite Borel measures $\nu$ and $\partial\nu$ that are absolutely continuous with respect to $\mu$ and $\partial\mu$, respectively, the $L^2(\nu)$ and $L^2(\partial\nu)$ losses

$$\int_K \|G(F(x)) - x\|^2 \, d\nu(x) \quad \text{and} \quad \int_{\partial K} \|G(F(x)) - x\|^2 \, d\partial\nu(x)$$

can be made arbitrarily small. To see this, first note that $G$ can be modified off of $F(K \setminus K_0)$ so that the modified $G$ maps $\mathbb{R}^k$ into the convex hull of $\{x \in \mathbb{R}^n : \text{dist}(x, K) < 2\varepsilon\}$, and the diameter of this convex hull is smaller than the diameter of $K$ plus $4\varepsilon$. Thus, the $L^\infty$ loss

$$\max_{x \in K} \|G(F(x)) - x\| < \text{diam } K + 4\varepsilon \tag{2}$$

is smaller than diam $K + 4\varepsilon$. This and (1) imply the pair of inequalities

$$\int_K \|G(F(x)) - x\|^2 \, d\nu(x) < (\text{diam } K + 4\varepsilon)^2 \nu(K_0) + \varepsilon^2 \nu(K),$$

$$\int_{\partial K} \|G(F(x)) - x\|^2 \, d\partial\nu(x) < (\text{diam } K + 4\varepsilon)^2 \partial\nu(K_0 \cap \partial K) + \varepsilon^2 \partial\nu(\partial K).$$

Since both right sides $\to 0$ as $\delta, \varepsilon \to 0$ by the same measure theory fact in Remark 7 (Folland, 1999, Thm 3.5), this establishes the claim. The claim seems interesting in part because the loss $\frac{1}{N}\sum_{i=1}^{N} \|G(F(x_i)) - x_i\|^2$ typically used to train autoencoders converges to the $L^2(\nu)$ loss as $N \to \infty$ with probability 1 under certain assumptions on the data $x_1, \ldots, x_N \in K$. Namely, convergence occurs if the data are drawn from a Borel probability measure $\nu$ and satisfy a strong law of large numbers, which occurs under fairly general assumptions on the data (they need not be independent) (Doob, 1990, Thm X.2.1), (Andrews, 1987, p. 1466), (Pötscher & Prucha, 1989, Thm 1, Thm 2). However, Theorem 2 in §4 implies that the $L^\infty$ loss (2) *cannot* be made arbitrarily small in general.

## 3 Numerical illustration

We next illustrate the results through the numerical learning of a deep neural network autoencoder. In our example, inputs and outputs of the network are three-dimensional, and the set $K$ is taken to be the union of two smoothly embedded submanifolds of $\mathbb{R}^3$. The first manifold is a unit circle centered at $x = y = 0$ and lying in the plane $z = 0$. The second manifold is a unit circle centered at $x = 1$, $z = 0$ and contained in the plane $y = 0$. See Figure 2 (left).

The choice of suitable neural net architecture "hyperparameters" (number of layers, number of units in each layer, activation function) is a bit of an art, since in theory just single-hidden layer architectures (with enough "hidden units" or "neurons") can approximate arbitrary continuous functions on compacts. After some experimentation, we settled on an architecture with three hidden layers of encoding with 128 units each, and similarly for the decoding layers. The activation functions are ReLU (Rectified Linear Unit) functions, except for the bottleneck and output layers, where we pick simply linear functions. Graphically this is shown in Figure 3. An appendix lists the Python code used for the implementation. We generated 500 points in each of the circles, and used 5000 epochs with a batch size of 20. We used Python's TensorFlow with Adaptive Moment Estimation (Adam) optimizer and a mean squared error loss function. The resulting decoded vectors are shown in Figure 2(right). Observe how the circles have been broken to make possible their embedding into $\mathbb{R}^1$.

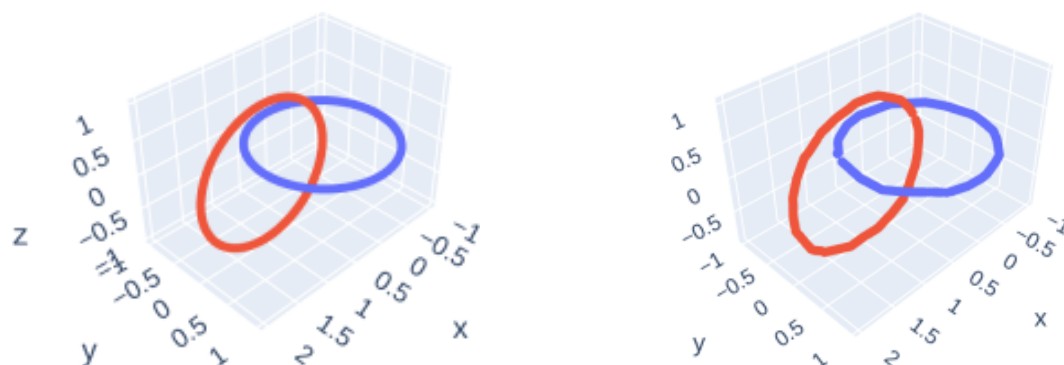

Figure 2: *Left:* Two interlaced unit circles, one centered at $x = y = 0$ in the plane $z = 0$ (blue), and another centered at $x = 1$, $z = 0$ in the plane $y = 0$ (red). The circles are parameterized as $x(\theta) = (\cos(\theta), \sin(\theta), 0)$ and $x(\theta) = (1 + \cos(\theta), 0, \sin(\theta))$ respectively, with $\theta \in [0, 2\pi]$. *Right:* The output of the autoencoder for the two interlaced unit circles, one centered at $x = y = 0$ in the plane $z = 0$ (blue), and another centered at $x = 1$, $z = 0$ in the plane $y = 0$ (red). The network learning algorithm automatically picked the points at which the circles should be "opened up" to avoid the topological obstruction.

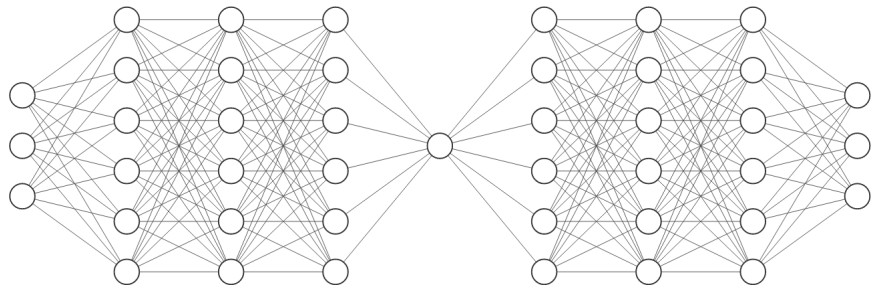

Figure 3: The architecture used in the computational example. For clarity in the illustration, only 6 units are depicted in each layer of the encoder and decoder, but the number used was 128.

The errors $\|G(F(x)) - x\|$ on the two circles are plotted in Figure 4. Observe that this error is relatively small except in two small regions.

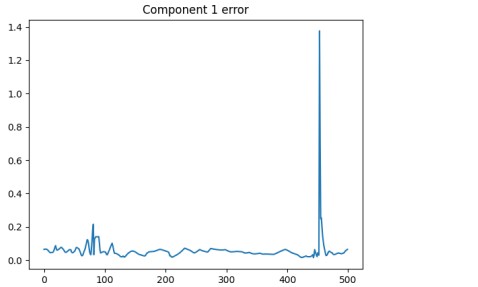 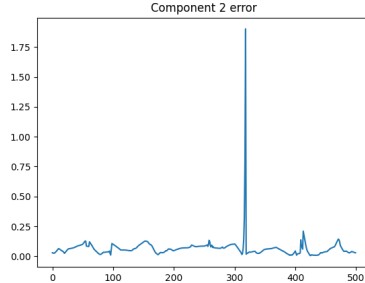

Figure 4: The errors $\|G(F(x)) - x\|$ on the two cirles. The $x$-axis shows the index $k$ representing the $k$th point in the respective circle, where $\theta = 2\pi k / 1000$.

In Figure 5 we show the image of the encoder layer mapping as a subset of $\mathbb{R}^1$ as well as the encoding map $F$.

It is important to observe that most neural net learning algorithms, including the one that we employed, are stochastic, and different executions might give different results or simply not converge. As an illustration of how results may differ, see Figures 6, 7, and 8.

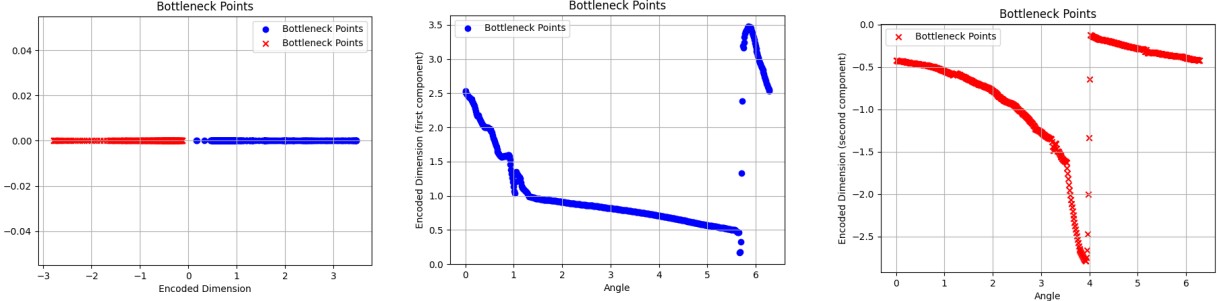

Figure 5: Left: The bottleneck layer, showing the images of the blue and red circles. Middle and Right: The encoding maps for the two circles. The $x$-axis is the angle $\theta$ in a $2\pi$ parametrization of the unit circles. The $y$-axis is the coordinate in the one-dimensional bottleneck layer.

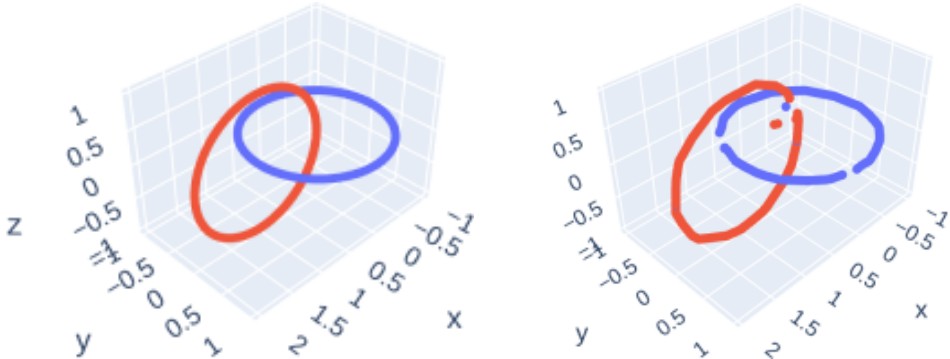

Figure 6: *Left:* Showing again the two interlaced unit circles. *Right:* For a different run of the algorithm, shown is the output of the autoencoder.

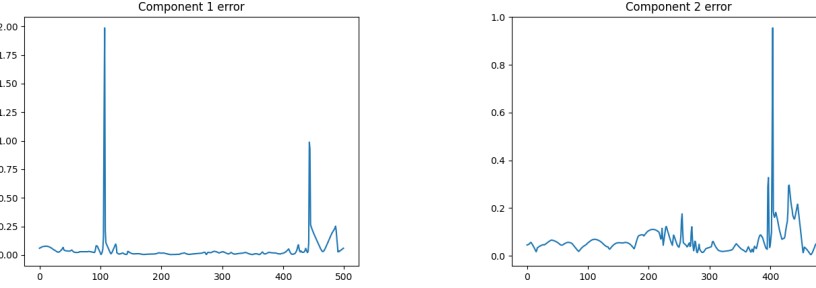

Figure 7: Result from another run of algorithm. The errors $\|G(F(x)) - x\|$ on the two circles. The $x$-axis shows the index $k$ representing the $k$th point in the respective circle, where $\theta = 2\pi k/500$.

## 4   Theorem 1 cannot be made global

Theorem 1 asserts that arbitrarily accurate autoencoding is always possible on the complement of a closed subset $K_0 \subseteq K$ having arbitrarily small positive intrinsic measure. This leads one to ask whether that result can be improved by imposing further "smallness" conditions on $K_0$. For example, rather than small positive measure, can one require that $K_0$ has measure zero? Alternatively, can one require that $K_0$ is small in the Baire sense, i.e., meager (§B.3)? In either case, the complement $K \setminus K_0$ of $K_0$ in $K$ would be dense, so the ability to arbitrarily accurately autoencode $K \setminus K_0$ as in Theorem 1 would imply the same for all of $K$. This is because continuity implies that the inequality (1) also holds with $K \setminus K_0$ replaced by its closure $\mathrm{cl}(K \setminus K_0)$, and $\mathrm{cl}(K \setminus K_0) = K$ if $K \setminus K_0$ is dense in $K$.

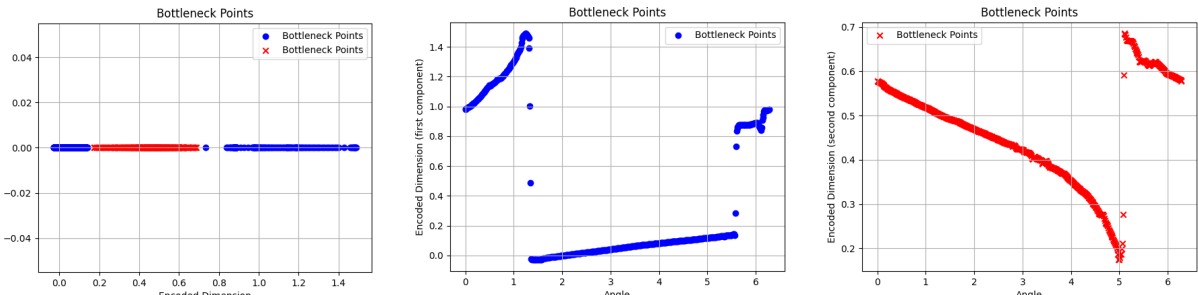

Figure 8: Result from another run of algorithm. Left: The bottleneck layer, showing the images of the blue and red circles. Middle and Right: The encoding maps for the two circles. The $x$-axis is the angle $\theta$ in a $2\pi$ parametrization of the unit circles. The $y$-axis is the coordinate in the one-dimensional bottleneck layer.

The following Theorem 2 eliminates the possibility of such extensions by showing that, for a broad class of $K$, the maximal autoencoder error on $K$ is bounded below by the **reach** $r_K \geq 0$ of $K$, a constant depending only on $K$. Here $r_K$ is defined to be the largest number such that any $x \in \mathbb{R}^n$ satisfying $\mathrm{dist}(x, K) < r_K$ has a unique nearest point on $K$ (Federer, 1959; Aamari et al., 2019; Berenfeld et al., 2022; Fefferman et al., 2016; 2018). Figure 9 illustrates this concept.

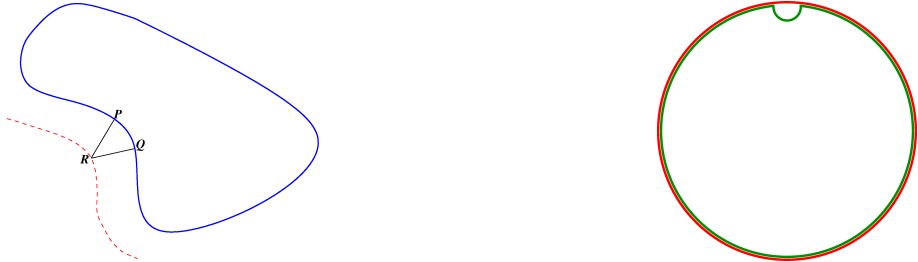

Figure 9: Left: Illustration of reach. A one-dimensional submanifold $K$ of $\mathbb{R}^2$ is shown in blue. Two segments are drawn normal to $K$, starting at points $P$ and $Q$ in a non-convex high-curvature region. These segments intersect at a point $R$ and have length $r_K$. If perturbations of $P$ and $Q$ lead to $R$, then there is no way to recover $P$ and $Q$ unambiguously as the unique point nearest to $R$. The dotted line represents points at distance $r_K$ from $K$. Right: Illustration of "dewrinkled" reach: here $K$ (in green) is a "dimpled circle" of radius 1, with a "dimple" which is a semicircle of radius $\varepsilon \approx 0$, and $L$ is the "ironed circle" of radius 1 in which the wrinkle has been removed. The mapping $T : L \to K$ is the obvious projection. In this example, $r_K = \varepsilon \approx 0$ but $r_{K,k}^* = 1 - \varepsilon \approx 1$.

**Remark 9.** The example $K := \{0\} \cup \{1/n \colon n \in \mathbb{N}\} \subseteq \mathbb{R}$ shows that a compact subset of a Euclidean space need not have a positive reach $r_K \geq 0$. However, $r_K > 0$ if $K$ is a compact smoothly embedded submanifold (cf. (3) below).

**Theorem 2.** Let $k, n \in \mathbb{N}$ and $K \subseteq \mathbb{R}^n$ be a $k$-dimensional compact smoothly embedded submanifold. For any continuous functions $F \colon \mathbb{R}^n \to \mathbb{R}^k$ and $G \colon \mathbb{R}^k \to \mathbb{R}^n$,

$$\max_{x \in K} \|G(F(x)) - x\| \geq r_K > 0. \tag{3}$$

**Remark 10.** The ability to make $K_0$ small in Theorem 1 relies on an autoencoder's ability to produce functions $G \circ F$ that change rapidly over small regions. E.g., if $G \circ F$ is Lipschitz then Theorem 2 implies a lower bound on the size of $K_0$ in terms of $r_K$ and the Lipschitz constant.

To prove Theorem 2 we instead prove the following more general Theorem 3, because the proof is the same. Here $H_k(S; \mathbb{Z}_2)$ denotes the $k$-th singular homology of a topological space $S$ with coefficients in the abelian group $\mathbb{Z}_2 := \mathbb{Z}/2\mathbb{Z}$ (Hatcher, 2002, p. 153). Upon taking $L = \mathbb{R}^k$ for the latent space, the statement implies

Theorem 2 since $H_k(K; \mathbb{Z}_2) = \mathbb{Z}_2 \neq 0$ when $K$ is a compact manifold (Hatcher, 2002, p. 236). Recall that $r_K$ denotes the reach of $K \subseteq \mathbb{R}^n$. See Appendix B (§B.4) for discussion of the topological concepts and results used in the following proof.

**Theorem 3.** Let $k, n \in \mathbb{N}$, $K \subseteq \mathbb{R}^n$ be a compact subset, and $L$ be a noncompact manifold of dimension less than or equal to $k$. If $H_k(K; \mathbb{Z}_2) \neq 0$, then for any continuous maps $F \colon K \to L$ and $G \colon L \to \mathbb{R}^n$,

$$\max_{x \in K} \|G(F(x)) - x\| \geq r_K. \tag{4}$$

*Proof.* Let $K \subseteq \mathbb{R}^n$ be a compact subset and $L$ be a noncompact manifold of dimension at most $k$. Since (4) holds automatically if $r_K = 0$, assume $r_K > 0$. We prove the contrapositive statement that failure of (4) for some $F, G$ implies that $H_k(K; \mathbb{Z}_2) = 0$. Thus, assume there are continuous maps $F, G$ such that

$$\max_{x \in K} \|G(F(x)) - x\| < r_K.$$

This implies that

$$G(F(K)) \subseteq N_{r_K}(K) := \{x \in \mathbb{R}^n \colon \operatorname{dist}(x, K) < r_K\}.$$

Since for each $x \in N_{r_K}(K)$ the optimization problem $\min_{y \in K} \operatorname{dist}(x, y)$ has a unique minimizer $y_* = \rho(x)$, $\rho \colon N_{r_K}(K) \to K$ is a continuous retraction ($\rho|_K = \operatorname{id}_K$). The line segment from $x \in K$ to $G(F(x))$ is contained in $N_{r_K}(K)$, since for $t \in [0, 1]$

$$\operatorname{dist}(tG(F(x)) + (1-t)x, K) \ \leq \ \|tG(F(x)) + (1-t)x - x\| \ \leq \ \|G(F(x)) - x\| \ < \ r_K.$$

Thus,

$$(t, x) \mapsto \rho\left(tG(F(x)) + (1-t)x\right)$$

defines a homotopy $[0, 1] \times K \to K$ from $\operatorname{id}_K$ to $(\rho \circ G \circ F)|_K \colon K \to K$. Defining the open set $U \subseteq \mathbb{R}^k$ containing $F(K)$ to be the preimage $U := G^{-1}(N_{r_K}(K))$, homotopy invariance (Hatcher, 2002, Thm 2.10, p. 153) implies that the induced homomorphism (Hatcher, 2002, p. 111)

$$(\rho \circ G \circ F|_K)_* = \rho_* \circ (G|_U)_* \circ F_* \colon H_k(K; \mathbb{Z}_2) \to H_k(K; \mathbb{Z}_2)$$

is equal to the identity homomorphism $(\operatorname{id}_K)_*$ induced by $\operatorname{id}_K$. On the other hand, the homomorphism

$$(G|_U)_* \colon H_k(U; \mathbb{Z}_2) \to H_k(N_{r_K}(K); \mathbb{Z}_2)$$

is zero, since $U$ is a noncompact manifold of dimension $\leq k$, and $H_k(U; \mathbb{Z}_2) = 0$ for any such $U$ (Hatcher, 2002, Prop. 3.29, Prop. 2.6). Thus, $H_k(K; \mathbb{Z}_2) = 0$. This completes the proof by contrapositive. $\qquad\square$

The reach is a globally defined parameter, and thus our lower bound on approximation error may underestimate the minimal possible error. In a private communication, Dr. Joshua Batson suggested that the authors consider an example such as the one shown in Figure 9(right) and attempt to prove a better lower bound for such an example, which led us to improve the necessary statement as follows.

For any two compact subsets $K$ and $L$ of $\mathbb{R}^n$, we denote by $\mathcal{C}(K, L)$ the set of continuous mappings $T \colon L \to K$, and define the maximum deviation of $T \in \mathcal{C}(K, L)$ from the identity as:

$$\delta(T) := \max_{y \in L} \|T(y) - y\|.$$

We denote by $\mathcal{M}_{n,k}$ the set of all compact smoothly embedded $k$-dimensional submanifolds $L$ of $\mathbb{R}^n$. For any compact subset $K \subseteq \mathbb{R}^n$, and any $k \in \mathbb{N}$, we define the $k$-dimensional **dewrinkled reach** as

$$r^*_{K,k} := \sup_{L \in \mathcal{M}_{n,k}, \, T \in \mathcal{C}(K,L)} \left\{r_L - \delta(T)\right\}.$$

When $K \in \mathcal{M}_{n,k}$, we have that $r^*_{K,k} \geq r_K$ (use $L = K$ and $T = \operatorname{identity}$). However, $r^*_{K,k}$ may be much larger than $r_K$ (see Figure 9(right)).

**Corollary 1.** Let $k, n \in \mathbb{N}$ and $K \subseteq \mathbb{R}^n$ a compact subset. For any continuous functions $F \colon \mathbb{R}^n \to \mathbb{R}^k$ and $G \colon \mathbb{R}^k \to \mathbb{R}^n$,

$$\max_{x \in K} \|G(F(x)) - x\| \geq r_{K,k}^* \,. \tag{5}$$

*Proof.* Pick $L \in \mathcal{M}_{n,k}$ and $T \in \mathcal{C}(K, L)$, and consider the composition $\widetilde{F} := F \circ T \colon L \to \mathbb{R}^k : y \mapsto F(T(y))$. Applying Theorem 3 to $L$ and the maps $\widetilde{F} \colon L \to \mathbb{R}^k$ and $G \colon \mathbb{R}^k \to \mathbb{R}^n$, we may pick a $\eta \in L$ so that $\|\eta - G(\widetilde{F}(\eta))\| \geq r_L$. Let $\xi := T(\eta)$, so $G(\widetilde{F}(\eta)) = G(F(T(\eta)) = G(F(\xi))$. Then

$$\eta - G(\widetilde{F}(\eta)) = \eta - G(F(\xi)) = (\eta - T(\eta)) + (\xi - G(F(\xi)))$$

so

$$r_L \leq \|\eta - G(\widetilde{F}(\eta))\| \leq \|\eta - T(\eta)\| + \|\xi - G(F(\xi))\| \leq \delta(T) + \|\xi - G(F(\xi))\|$$

and hence

$$\max_{x \in K} \|G(F(x)) - x\| \geq \|\xi - G(F(\xi))\| \geq r_L - \delta(T) \,.$$

This is valid for all $(L, T)$, and thus $\max_{x \in K} \|G(F(x)) - x\| \geq r_{K,k}^*$, as claimed. $\square$

**Remark 11.** All the results in this section were stated for manifolds, meaning (recall our convention) manifolds with empty boundary. Clearly, the same results cannot be valid for manifolds with non-empty boundary. For example, the submanifold with boundary of $\mathbb{R}^2$ consisting of a one-dimensional segment in the $x$-axis has infinite reach yet can be perfectly reconstructed (project on $x$-axis and then include in $\mathbb{R}^2$).

## 5 Discussion

Our main representation result is Theorem 1. This theorem theoretically insures that data points lying in a submanifold $K$ (or even in a finite union of submanifolds) of a given dimension $k$ can be encoded through a bottleneck layer of the same dimension $k$, up to an arbitrarily small uniform reconstruction error $\varepsilon$. Moreover, the generalization error will also be uniformly smaller than $\varepsilon$, with arbitrarily high probability $1 - \delta$, when points are randomly sampled from $K$. Our main necessity result is Theorem 2. This theorem complements the representability result by providing a lower bound for global uniform reconstruction. On the other hand, as discussed in Remark 8, one can guarantee a global reconstruction with error less than $\varepsilon$ in a mean least squares sense.

There is a vast amount of experimental work using autoencoders for dimension reduction, but comparatively few papers focus on a theoretical basis for such reductions. One theoretical result is given (with no proof) in Hecht-Nielsen (1995), in which a theorem is stated for replicator neural networks (with quantized middle hidden layer activations approximating the function $\theta(r) = 0$ for $r < 0$, $\theta(r) = 1$ for $r > 1$ and $\theta(r) = r$ for $r \in [0, 1]$). Using our notation, the theorem claims roughly that if data belongs to a set $K$ which is the image of a smooth embedding of a $k$-dimensional unit cube, and a probability measure is given on $K$, then, in the limit of high dimensions ($k \to \infty$) and a large number of quantization levels, replicator networks trained to compute optimal encodings will recover the natural (entropy) coordinates in the data manifold. Our Theorem 1, in contrast, studies representations of data lying in rather arbitrary manifolds (and would indeed be quite trivial if $K$ was already assumed to be diffeomorphic to a cube), and is valid for arbitrary $k$, not merely asymptotically.

Regarding the limitations of autoencoders as reflected in our lower bounds for global reconstruction, the authors of Batson et al. (2021), in the context of anomaly detection in high-energy physics, argue that autoencoders might miss or falsely detect anomalies due to the topological shape of the phase space. Our necessity result Theorem 2 serves to quantify these obstructions.

Theorem 1 provides an existence result. As is often the case with results regarding the expressive power of neural networks, effective learning during training involves overcoming numerous challenges. This is because the landscape of the loss function (whether $L^2$ or any other criterion) is typically highly non-convex and irregular, presenting spurious local minima, plateaus, and potentially steep ravines, leading gradient-based optimization methods to converge to local minima or navigate through saddle points inefficiently, thus

failing to find a low-error autoencoder. Moreover, the choice of optimizer and network architecture and hyperparameters will affect the success of numerical methods. Finally, for sparse training samples from $K$ there is little hope of effective generalization to the full manifold $K$ in the absence of proper regularization of the loss function.

There are many possible directions in which we will be expanding our study. One of them is the extension to model reduction for time series data, for which there are many existing approaches including for example dynamic mode decomposition (Kutz et al., 2016) and deep learning dynamic mode decomposition (Alford-Lago et al., 2022). Specifically, one may assume that a vector field, or an iteration in discrete-time, exists on the data manifold $K$. The objective then becomes that of defining a dynamics in the bottleneck layer that intertwines with the original dynamics in $K$, thus providing a reduced-order representation of the original dynamics, in the spirit of the computational approach in Baig et al. (2023). Further along this direction, one may consider control systems (thought of as families of vector fields), and the reduction to lower-dimensional control problems in the same fashion.

A related direction of study concerns representation of dynamics through the "Koopman" approach, in which the middle-layer dynamics are linearized. Theoretical results characterizing the limitations as well as possibilities of Koopman embeddings are given in Liu et al. (2023); Kvalheim & Arathoon (2023). In this context, the middle dimension is often larger than the input dimension, rather than smaller, but on the other hand linearity imposes a different type of simplification. Autoencoder realizations of Koopman embeddings have been suggested in the literature, see for instance Otto & Rowley (2019); Azencot et al. (2020). We will extend the theory to establish when Koopman autoencoders exist, and their limitations. In parallel or in combination with these dynamics ideas, if our manifold $K$ comes endowed with a particular probability measure, we may ask to represent this measure through the bottleneck, as a distribution on latent variables, which is a topic closely related to variational autoencoders.

Yet another direction of research is that of understanding to what extent latent representations can mirror, or not, global topological, metric, and combinatorial features of data manifolds, adapting and extending the recent work Wang et al. (2023) that dealt with the unavoidable distortions that arise from low-dimensional representations, especially in the context of systems biology single cell data.

Finally, another direction of study concerns the generalization of our representation result Theorem 1 from unions $K$ of submanifolds with boundary to unions of more general stratified sets (Trotman, 2020, Def. 1.11). A manifold with boundary is an example of a stratified set with two strata, namely, the codimension-0 interior and codimension-1 boundary. Most of the conclusions of Theorem 1 are "stratified" in the sense that the conclusion $\partial\mu(K_0 \cap \partial K) < \delta$ for the codimension-1 stratum is the analog of the conclusion $\mu(K_0) < \delta$ for the codimension-0 stratum, and the conclusion (1) directly implies the analogous conclusion with $K$ replaced by $\partial K$. However, Theorem 1 contains no statement on connectedness of $(\partial M) \setminus K_0$ analogous to the conclusion of Theorem 1 that $M \setminus K_0$ is connected for each component $M$ of $K$. It seems interesting to know whether this analogous statement generally holds, and moreover whether Theorem 1 generalizes to a useful class of stratified sets in a "fully stratified" way. For example, a suitable generalization of Theorem 1 to Whitney stratified sets (Trotman, 2020, Def. 1.2.3) would imply a representation theorem for autoencoding of algebraic varieties and more generally subanalytic sets, since these admit Whitney stratifications (Trotman, 2020, p. 5). Algebraic varieties arise naturally as the sets of steady states of mass-action biological systems, and finding parametrizations of steady states is a key problem in fitting models to data. In the special case of varieties defined by toric ideals, global parametrizations are possible (Chaves et al., 2004), but in more general cases, particularly when analyzing single-cell data, equilibrium sets are only known numerically (Wang et al., 2019), and autoencoders might provide a useful approach to the estimation of dimension.

### Acknowledgments

EDS's work was partially supported by grants ONR N00014-21-1-2431 and AFOSR FA9550-21-1-0289. The authors thank the reviewers for insightful comments, and especially thank Dr. Joshua Batson for his observations that led to Remark 8 and to Corollary 1.

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

## A Appendix: Code used for implementation

```
howmany_points = 500
epochs = 5000
batch_size = 20
import matplotlib.pyplot as plt
import plotly.graph_objects as go
import pandas as pd
import numpy as np
import scipy as sp
import tensorflow as tf
from tensorflow.keras.layers import Input, Dense
from tensorflow.keras.models import Model

# Define the parametric equations for the circles
def circle_xy(t, h, k, r):
    x = h + r * np.cos(t)
    y = k + r * np.sin(t)
    z = 0 * np.ones_like(t)
    return x, y, z

def circle_yz(t, h, k, r):
    x = h + r * np.sin(t)
    y = 0 * np.ones_like(t)
    z = k + r * np.cos(t)
    return x, y, z

t = np.linspace(0, 2 * np.pi, howmany_points)

x1, y1, z1 = circle_xy(t, 0, 0, 1)
x2, y2, z2 = circle_yz(t, 1, 0, 1)

input_data = np.vstack((np.column_stack((x1, y1, z1)),\
np.column_stack((x2, y2, z2))))
# Build the autoencoder architecture with a bottleneck layer of dimension 1
input_data_test = np.vstack((np.column_stack((x1test, y1test, z1test)),\
np.column_stack((x2test, y2test, z2test))))

input_dim = 3

# Encoder model
input_layer = Input(shape=(input_dim,))
encoded = Dense(128, activation='relu')(input_layer)
encoded = Dense(128, activation='relu')(encoded)
encoded = Dense(128, activation='relu')(encoded)
encoded = Dense(1, activation='linear')(encoded)  # Bottleneck layer with dimension 1
encoder = Model(inputs=input_layer, outputs=encoded)

# Decoder model
decoded_input = Input(shape=(1,))
decoded = Dense(128, activation='relu')(decoded_input)
decoded = Dense(128, activation='relu')(decoded)
decoded = Dense(128, activation='relu')(decoded)
decoded = Dense(input_dim, activation='linear')(decoded)
decoder = Model(inputs=decoded_input, outputs=decoded)

# Autoencoder model
autoencoder = Model(inputs=input_layer, outputs=decoder(encoder(input_layer)))
```

```
autoencoder.compile(optimizer='adam', loss='mean_squared_error')

autoencoder.fit(input_data, input_data, epochs=epochs, \
batch_size=batch_size, shuffle=True)

# Test the autoencoder on the training data
encoded_vectors = encoder.predict(input_data)
decoded_vectors = decoder.predict(encoded_vectors)

decoded_vectors_1 = decoded_vectors[0:howmany_points,:]
decoded_vectors_2 = decoded_vectors[-howmany_points:,:]
encoded_vectors_1 = encoded_vectors[0:howmany_points,:]
encoded_vectors_2 = encoded_vectors[-howmany_points:,:]

# Create the 3D plot of data vectors in plotly
fig1 = go.Figure()
# Add circles to the plot
fig1.add_trace(go.Scatter3d(x=x1, y=y1, z=z1, mode='lines',\
name='unit circle centered at x=0, y=0 in the plane z=0', line=dict(width=8)))
fig1.add_trace(go.Scatter3d(x=x2, y=y2, z=z2, mode='lines',\
name='unit circle centered at x=1, z=0 in the plane y=0', line=dict(width=8)))
# Setting the axis labels
zoom = 2.5
fig1.update_layout(scene_camera=dict(eye=dict(x=zoom, y=zoom, z=zoom)))\
# zoom out so plot fits
fig1.show()
fig1.write_image(pathdrive+"original.png") #this works with plotly
fig1.write_image(pathdrive+"original.svg")

fig2 = go.Figure()
fig2.add_trace(go.Scatter3d(x=decoded_vectors_1[:, 0], y=decoded_vectors_1[:, 1], z=decoded_vectors_1[:,2],\
mode='markers', marker=dict(size=3), name='decoded unit circle centered at x=0, y=0 in the plane z=0'))
fig2.add_trace(go.Scatter3d(x=decoded_vectors_2[:, 0], y=decoded_vectors_2[:, 1], z=decoded_vectors_2[:,2],\
mode='markers', marker=dict(size=3), name='decoded unit circle centered at x=0, y=0 in the plane z=0'))
zoom = 2
fig2.update_layout(scene_camera=dict(eye=dict(x=zoom, y=zoom, z=zoom))) # zoom out so plot fits
fig2.show()
fig2.write_image(pathdrive+"decoded1.png") #this works with plotly
fig2.write_image(pathdrive+"decoded1.svg")

# Create again the 3D plot of data vectors in plotly but use a different view in 3 and 4 below:
fig3 = go.Figure()
# Add circles to the plot
fig3.add_trace(go.Scatter3d(x=x1, y=y1, z=z1, mode='lines',\
name='unit circle centered at x=0, y=0 in the plane z=0',\
line=dict(width=8)))
fig3.add_trace(go.Scatter3d(x=x2, y=y2, z=z2, mode='lines',\
name='unit circle centered at x=1, z=0 in the plane y=0', line=dict(width=8)))
# Setting the axis labels
fig3.update_layout(scene=dict(xaxis_title='X', yaxis_title='Y',\
zaxis_title='Z'))
# to convert spherical elev=30, azim=65 to cartesian, one uses
#      x = r * math.cos(elev_rad) * math.cos(azim_rad)
#      y = r * math.cos(elev_rad) * math.sin(azim_rad)
#      z = r * math.sin(elev_rad)
# so I get with r=1: x=0.366, y=0.785, z=0.5
zoom = 3
fig3.update_layout(scene_camera=dict(eye=dict(x=zoom*0.366, y=zoom*0.785,
z=zoom*0.5)))
```

```
# different view angle zoom out so plot fits
fig3.show()
fig3.write_image(pathdrive+"original2.png") #this works with plotly
fig3.write_image(pathdrive+"original2.svg")

fig4 = go.Figure()
fig4.add_trace(go.Scatter3d(x=decoded_vectors_1[:, 0], y=decoded_vectors_1[:,
1],\
z=decoded_vectors_1[:,2], mode='markers', marker=dict(size=3),\
name='decoded unit circle centered at x=0, y=0 in the plane z=0'))
fig4.add_trace(go.Scatter3d(x=decoded_vectors_2[:, 0], y=decoded_vectors_2[:,
1],\
z=decoded_vectors_2[:,2], mode='markers', marker=dict(size=3),\
name='decoded unit circle centered at x=0, y=0 in the plane z=0'))
fig4.update_layout(scene=dict(xaxis_title='X', yaxis_title='Y', zaxis_title='Z'))
zoom = 3
fig4.update_layout(scene_camera=dict(eye=dict(x=zoom*0.366, y=zoom*0.785,\
z=zoom*0.5))) # zoom out so plot fits
fig4.show()
fig4.write_image(pathdrive+"decoded2.png") #this works with plotly
fig4.write_image(pathdrive+"decoded2.svg")

# Plot the bottleneck points
plt.scatter(encoded_vectors_1, np.zeros_like(encoded_vectors_1),\
marker='o', label='Bottleneck Points', color='b')
plt.scatter(encoded_vectors_2, np.zeros_like(encoded_vectors_1),\
marker='x', label='Bottleneck Points', color='r')
plt.xlabel('Encoded Dimension')
plt.title('Bottleneck Points')
plt.legend()
plt.grid()
plt.savefig(pathdrive+"bottleneck.png") # this works with matplotlib but before show
plt.tight_layout()
plt.show()

# compute matrix norm along second "axis", i.e. along "y axis", i.e. each row
delta_1 = np.linalg.norm(input_data[0:howmany_points,:] - decoded_vectors_1, axis = 1)
delta_2 = np.linalg.norm(input_data[-howmany_points:,:] - decoded_vectors_2, axis = 1)

plt.plot(delta_1)
plt.title('Component 1 error')
plt.savefig(pathdrive+"error1.png")
# this works with matplotlib but before show
plt.show()

plt.plot(delta_2)
plt.title('Component 2 error')
plt.savefig(pathdrive+"error2.png")
plt.show()

# plot the encoded as a function of the angle parameter
plt.scatter(t, encoded_vectors_1, marker='o', label='Bottleneck Points', color='b')
plt.xlabel('Angle')
plt.ylabel('Encoded Dimension (first component)')
plt.title('Bottleneck Points')
plt.legend()
plt.grid()
plt.savefig(pathdrive+"encoding1.png")
plt.show()
```

```
plt.scatter(t, encoded_vectors_2, marker='x', label='Bottleneck Points', color='r')
plt.xlabel('Angle')
plt.ylabel('Encoded Dimension (second component)')
plt.title('Bottleneck Points')
plt.legend()
plt.grid()
plt.savefig(pathdrive+"encoding2.png")
plt.show()
```

# B    Appendix: Review of some basic concepts and results in topology

In this appendix we review basic concepts and results in topology that are used in this paper. We discuss general topology in §B.1, finite Borel measures in §B.2, differential topology in §B.3, and algebraic topology in §B.4.

## B.1    General topology

A **topology** on a set $X$ is a collection of subsets of $X$, called **open**, satisfying the following three properties (Lee, 2013, p. 596):

- $X$ and $\varnothing$ are open.

- The union of any family of open sets is open.

- The intersection of any finite family of open sets is open.

A set $X$ equipped with a topology is called a **topological space**.

A subset $C \subseteq X$ is **closed** if its complement $X \setminus C$ is open (Lee, 2013, p. 596). The **closure** $\mathrm{cl}(S)$ of a subset $S \subseteq X$ of a topological space $X$ is the intersection of all closed sets containing $S$ (Lee, 2013, p. 597). Thus, $S \subseteq X$ is closed if and only if $\mathrm{cl}(S) = S$. A subset $S \subseteq X$ is **dense** if $\mathrm{cl}(S) = X$.

A topological space $X$ is **connected** if it is not the union of any two disjoint non-empty open sets (Lee, 2013, p. 607). A topological space $X$ is **compact** if, for any collection of open sets whose union is $X$, there is a finite subcollection whose union is $X$ (Lee, 2013, p. 608).

Given a subset $S \subseteq X$ of a topological space, the **subspace topology** is the topology on $S$ that declares a subset $U \subseteq S$ to be open in $S$ if and only if there is a subset $V \subseteq X$ open in $X$ such that $U = V \cap S$ (Lee, 2013, p. 601). A subset $S \subseteq X$ is **connected** if it is connected in the subspace topology, and **compact** if it is compact in the subspace topology (Lee, 2013, pp. 607–608). A **(connected) component** of $X$ is a connected subset of $X$ that is not a proper subset of any larger connected subset (Lee, 2013, p. 607).

A topological space $X$ is **Hausdorff** if any pair of distinct points in $X$ are contained in some pair of disjoint open sets, and is **second-countable** if there is a countable collection of open sets such that every open set in $X$ is a union of some open sets from the countable collection (Lee, 2013, p. 600). Every subset of a Hausdorff space is Hausdorff in the subspace topology, and every subset of a second-countable space is second-countable in the subspace topology (Lee, 2013, Prop. A.17). Only second-countable Hausdorff topological spaces appear in the body of this paper.

**Example 1** (Lee (2013, Ex. A.6))**.** The standard topology on Euclidean space $\mathbb{R}^n$ is defined as follows. A subset $U \subseteq \mathbb{R}^n$ is declared to be open if for each point $x \in U$ there is some $r > 0$ such that the ball $N_r(x) \coloneqq \{y \in \mathbb{R}^n \colon \|x - y\| < r\}$ is a subset of $U$. These open sets can be checked to satisfy the three properties above, so they define a topology on $\mathbb{R}^n$. This topology is Hausdorff since any pair of points are contained in disjoint balls with positive radii, and is second-countable, as follows from the fact that every real number may be approximated by rational numbers. The **Heine-Borel** theorem asserts that a subset of $\mathbb{R}^n$ is compact if and only if it is closed and has bounded diameter (Lee, 2013, p. 608).

A map $F \colon X \to Y$ between topological spaces is **continuous** if the **preimage**

$$F^{-1}(U) \coloneqq \{x \in X \colon F(x) \in U\}$$

of any open subset of $Y$ is open in $X$ (Lee, 2013, p. 597). A bijective continuous map $F \colon X \to Y$ is a **homeomorphism** if the inverse map $F^{-1} \colon Y \to X$ is continuous (Lee, 2013, p. 597). An injective continuous map $F \colon X \to Y$ is a **topological embedding** if the codomain-restricted map $F \colon X \to F(X)$ is a homeomorphism when the **image**

$$F(X) \coloneqq \{F(x) \colon x \in X\} \subseteq Y$$

of $F$ is given the subspace topology inherited from $Y$ (Lee, 2013, p. 601).

The **product topology** on the Cartesian product $X \times Y$ of topological spaces $X$ and $Y$ is defined by declaring a subset $S \subseteq X \times Y$ to be open if, for each $(x, y) \in S$, there are open sets $U \subseteq X$ and $V \subseteq Y$ respectively containing $x$ and $y$ such that $U \times V \subseteq S$.

### B.2 Finite Borel measures

A subset $S \subseteq X$ of a topological space $X$ is a **Borel set** if it can be formed from open subsets via the operations of taking countable unions, taking countable intersections, and taking complements within $X$ (Folland, 1999, p. 22). A **finite Borel measure** $\mu$ on $X$ is a map from the Borel sets to the nonnegative real numbers such that $\mu(\varnothing) = 0$ and $\mu(\bigcup_{j=1}^{\infty} S_j) = \sum_{j=1}^{\infty} \mu(S_j)$ for any countable family of pairwise disjoint Borel sets $S_1, S_2, \ldots \subseteq X$ (Folland, 1999, pp. 24–25). A finite Borel measure $\mu$ on $X$ is a **probability measure** if $\mu(X) = 1$.

A map $F \colon X \to Y$ between topological spaces is **Borel measurable** if $F^{-1}(S)$ is a Borel set in $X$ for any Borel set $S$ in $Y$. For any Borel measurable function $f \colon X \to [0, \infty)$ and finite Borel measure $\mu$ on $X$, there is a well-defined integral $\int_X f(x) \, d\mu(x) \in [0, \infty]$ (Folland, 1999, p. 50).

A finite Borel measure $\nu$ on $X$ is **absolutely continuous** with respect to a finite Borel measure $\mu$ on $X$ if $\nu(S) = 0$ whenever $\mu(S) = 0$ (Folland, 1999, p. 88). In this case, the **Radon-Nikodym theorem** asserts the existence of a Borel measurable function $f \colon X \to [0, \infty)$ such that

$$\nu(S) = \int_S f(x) \, d\mu(x) \coloneqq \int_X \mathbf{1}_S(x) f(x) \, d\mu(x)$$

for each Borel set $S$, where $\mathbf{1}_S(x) = 1$ if $x \in S$ and $\mathbf{1}_S(x) = 0$ otherwise (Folland, 1999, p. 91).

A finite Borel measure $\mu$ on $X$ is **outer regular** if

$$\mu(S) = \inf\{\mu(U) \colon U \supseteq S, U \text{ is open}\}$$

for all Borel sets $S \subseteq X$ (Folland, 1999, p. 212).

### B.3 Differential topology

A topological space $M$ is an $n$**-dimensional (topological) manifold with boundary** if it is second-countable, Hausdorff, and for each point $x \in M$ there is an open set $U \subseteq M$ containing $x$ that is homeomorphic to an open subset (with the subspace topology) of the **closed $n$-dimensional upper half-space** (Lee, 2013, p. 25)

$$\mathbb{H}^n \coloneqq \{(x_1, \ldots, x_n) \in \mathbb{R}^n \colon x_n \geq 0\}.$$

A choice of homeomorphism $\varphi \colon U \to \varphi(U) \subseteq \mathbb{H}^n$ is called a **chart** $(U, \varphi)$ for $M$. We say that the chart $(U, \varphi)$ **contains** the point $x \in M$ if $x \in U$. A point $x \in M$ is called an **interior point** if the $n$-th coordinate of $\varphi(x) \in \mathbb{H}^n$ is positive for some chart $(U, \varphi)$ containing $x$, and a **boundary point** otherwise. The collection of boundary points is called the **(manifold) boundary** of $M$, denoted by $\partial M$, and the complement $\mathrm{int}(M) \coloneqq M \backslash \partial M$ is called the **(manifold) interior** of $M$. We say that $M$ is an $n$**-dimensional (topological) manifold** if $\partial M = \varnothing$. (Equivalently, one can define $n$-dimensional manifolds by replacing $\mathbb{H}^n$ by $\mathbb{R}^n$ in the definition of $n$-dimensional manifolds with boundary (Lee, 2013, pp. 2–3).)

A map between open subsets of Euclidean spaces is **smooth** if it has continuous partial derivatives of all orders. Given an arbitrary subset $A \subseteq \mathbb{R}^n$, a map $F \colon A \to \mathbb{R}^m$ is **smooth** if for each $x \in A$ there is an open set $U \subseteq \mathbb{R}^n$ and a smooth map $\tilde{F} \colon U \to \mathbb{R}^m$ whose restriction $\tilde{F}|_{U \cap A}$ coincides with $F|_{U \cap A}$ (Lee, 2013, p. 645). Given a subset $B \subseteq \mathbb{R}^m$, we say that $F \colon A \to B$ is **smooth** if $F$ is smooth when viewed as a map into $\mathbb{R}^m$.

Let $M$ be an $n$-dimensional manifold with boundary. Two charts $(U, \varphi)$, $(V, \psi)$ are called **smoothly compatible** if either $U \cap V = \varnothing$ or the **transition map** $\psi \circ \varphi^{-1} \colon \varphi(U \cap V) \to \psi(U \cap V) \subseteq \mathbb{R}^n$ is smooth (in the

sense of the previous paragraph). A **smooth atlas** for $M$ is a collection of smoothly compatible charts such that the union of chart domains is $M$. A smooth atlas for $M$ is **maximal** if it is not properly contained in any larger smooth atlas. A **smooth structure** on $M$ is a maximal smooth atlas (Lee, 2013, p. 28).

An $n$-**dimensional smooth manifold with boundary** is an $n$-dimensional manifold with boundary $M$ equipped with a choice of smooth structure (Lee, 2013, p. 28). Such an $M$ is an $n$-**dimensional smooth manifold** if $\partial M = \varnothing$. (Equivalently, one can define $n$-dimensional smooth manifolds by replacing $\mathbb{H}^n$ by $\mathbb{R}^n$ in the definition of $n$-dimensional smooth manifolds with boundary (Lee, 2013, pp. 4, 12–13).)

**Example 2.** Euclidean space $\mathbb{R}^n$ is an $n$-dimensional manifold. Every $x \in \mathbb{R}^n$ is contained in the domain of the chart $(\mathbb{R}^n, \mathrm{id}_{\mathbb{R}^n})$ defined by the identity map. The union of this chart with all charts smoothly compatible with it defines the standard smooth structure on $\mathbb{R}^n$ making it a smooth manifold (Lee, 2013, Ex 1.22). Similarly, $\mathbb{H}^n$ is an $n$-dimensional manifold with boundary, and a smooth manifold with boundary when equipped with the standard smooth structure consisting of of all charts smoothly compatible with $(\mathbb{H}^n, \mathrm{id}_{\mathbb{H}^n})$.

Let $M$, $N$ be smooth manifolds with boundary and $A$ be an arbitrary subset of $M$. A map $F \colon A \to N$ is **smooth** if for each $x \in A$ there is a chart $(U, \varphi)$ containing $x$ and a chart $(V, \psi)$ containing $F(x)$ such that $F(U) \subseteq V$ and $\psi \circ F \circ \varphi^{-1} \colon \varphi(U \cap A) \to \psi(V)$ is a smooth map between subsets of Euclidean spaces in the sense defined above (Lee, 2013, p. 45), (Lee, 2024, p. 1). When $N = \mathbb{R}^n$ and $A$ is closed, such an $F$ always admits a **smooth extension** $\tilde{F} \colon M \to \mathbb{R}^n$, meaning that $\tilde{F}$ is smooth and $\tilde{F}|_A = F$ (Lee, 2013, Lem. 2.26).

A smooth map $F \colon M \to N$ is a **smooth embedding** if it is a topological embedding and the inverse $F^{-1} \colon F(M) \to N$ is smooth. (This is equivalent to the usual definition (Lee, 2013, p. 85) by the chain rule (Lee, 2013, Prop. 3.6(b))). A **diffeomorphism** is a bijective smooth embedding (Lee, 2013, p. 38).

Let $x$ be a point in an $n$-dimensional smooth manifold with boundary $M$, and consider smooth curves $\gamma \colon J_\gamma \to M$ that are defined on some interval $J_\gamma \subseteq \mathbb{R}$ containing $0$ and satisfy $\gamma(0) = x$. A **tangent vector** at $x \in M$ is an equivalence class of such curves, where curves $\gamma_1$, $\gamma_2$ are called equivalent if $\frac{d}{dt}\varphi(\gamma_1(t))|_{t=0} = \frac{d}{dt}\varphi(\gamma_2(t))|_{t=0}$ for some smooth chart $(U, \varphi)$ containing $x$ (Lee, 2013, pp. 70, 72). The **tangent space** $T_x M$ at $x \in M$ is an $n$-dimensional vector space that consists of all tangent vectors at $x$. The **tangent bundle** of $M$ is the disjoint union $TM := \bigsqcup_{x \in M} T_x M$ of all tangent spaces, and it has a canonical topology and smooth structure making it into a $2n$-dimensional smooth manifold with boundary (Lee, 2013, pp. 66–67).

A **smooth vector field** $Y$ on a smooth manifold with boundary $M$ is a smooth map $Y \colon M \to TM$ satisfying $Y(x) \in T_x M$ for each $x \in M$ (Lee, 2013, p. 175). A point $p \in M$ such that $Y(p) = 0$ is called an **equilibrium** (or **zero**) of $Y$. A smooth vector field is **inward-pointing** if for each $x \in \partial M$ there is a curve in the equivalence class defining $Y(x)$ that is defined on an interval of the form $[0, \varepsilon)$ (Lee, 2013, p. 118).

When $M$ is compact, an inward-pointing smooth vector field $Y$ on $M$ canonically determines a smooth map $\Phi \colon [0, \infty) \times M \to M$ such that the time-$t$ maps $\Phi^t := \Phi(t, \cdot)$ are (dimension-preserving) smooth embeddings satisfying $\Phi^0 = \mathrm{id}_M$ and $\Phi^{t+s} = \Phi^t \circ \Phi^s$ for all $t, s \geq 0$. This **semiflow** $\Phi$ is the unique such map with the property that each **trajectory** $t \mapsto \Phi^{t+s}(x)$ belongs to the equivalence class $Y(\Phi^s(x))$. (One constructs $\Phi$ by repeating, mutatis mutandis, the proof of Lee (2013, Thm 9.16) for the case $\partial M = \varnothing$; cf. Lee (2013, Thm 9.34)).

When $p \in M$ is an equilibrium of an inward-pointing smooth vector field $Y$ with solution map $\Phi$, $\Phi^t(p) = p$ for all $t \geq 0$. The equilibrium $p$ is called **hyperbolic** if none of the eigenvalues of the Jacobian matrix $D(\varphi \circ \Phi^1 \circ \varphi^{-1})(\varphi(p))$ have complex modulus equal to 1, where $(U, \varphi)$ is a chart containing $p$. The equilibrium $p$ is called **asymptotically stable** if for every open set $V \subseteq M$ containing $p$ there is an open set $U \subseteq V$ containing $p$ such that, for each $q \in U$, the the trajectory $t \mapsto \Phi^t(q)$ takes values in $V$ and converges to $p$ as $t \to \infty$ (Pajitnov, 2006, p. 74). The **basin of attraction** of an asymptotically stable equilibrium $p \in M$ is a connected open set consisting of all $q \in M$ such that the trajectory $t \mapsto \Phi^t(q)$ converges to $p$.

A **Riemannian metric** $g$ on a smooth manifold with boundary $M$ is an inner product $(Y_x, Z_x) \mapsto g(Y_x, Z_x)$ on each tangent space $T_x M$ such that $x \mapsto g(Y(x), Z(x))$ is a smooth map for any smooth vector fields $Y$, $Z$ on $M$ (Lee, 2013, Prop. 12.19, pp. 327–328). In particular, a Riemannian metric determines a smooth

**gradient** vector field $\nabla\varphi$ for each smooth function $\varphi\colon M \to \mathbb{R}$ (Lee, 2013, p. 342). If $\varphi\colon M \to [0,1]$ is smooth and $\partial M = \varphi^{-1}(1)$, then $\nabla\varphi$ is inward-pointing.

A Riemannian metric on a compact smooth manifold with boundary $M$ also determines an entity, called the **Riemannian density** (Lee, 2013, Prop. 16.45), that can be integrated over Borel measurable subsets of $M$ (cf. Lee (2013, p. 431)) to define a finite Borel measure $\mu$ on $M$ that is outer regular (§B.2), Folland (1999, Thm 7.8). A Borel set $A \subseteq M$ is called **measure zero** if $\mu(A) = 0$. By construction, changing the Riemannian metric changes $\mu$ to a finite Borel measure $\nu$ such that $\mu$, $\nu$ are absolutely continuous with respect to each other, so the property of being measure zero is well-defined independent of the choice of Riemannian metric. Alternatively, one can define "measure zero" without referring to any Riemannian metric (Lee, 2013, p. 128). If $F\colon M \to N$ is a smooth map between $n$-dimensional smooth manifolds with boundary and $A \subseteq M$ has measure zero, then $F(A) \subseteq N$ is also measure zero (Lee, 2013, Thm 5.9).

"Measure zero" provides one notion of what it means for a subset of a smooth manifold with boundary $M$ to be "small". An alternative topological "smallness" notion for subsets is "meager" . A subset $S \subseteq M$ is **nowhere dense** if $M \setminus \operatorname{cl}(S)$ is dense, and is **meager** if it is a countable union of nowhere dense sets (Folland, 1999, p. 161). The **Baire category theorem** asserts that the complement $M \setminus S$ of any meager set $S$ is dense (Lee, 2013, Thm A.58), (Folland, 1999, Thm 5.9).

A **diffeotopy** (or **ambient isotopy**) of a smooth manifold with boundary $M$ is a smooth map $J\colon [0,1]\times M \to M$ such that each time-$t$ map $J_t \coloneqq J(t,\cdot)$ is a diffeomorphism and $J_0 = \operatorname{id}_M$ (Hirsch, 1994, p. 178). The **support** of a diffeotopy $J$ of $M$ is the closure in $M$ of the set

$$\{x \in M\colon J_t(x) \neq x \text{ for some } t \in [0,1]\}.$$

Given a diffeotopy $\partial J$ of $\partial M$ and a diffeotopy $\tilde{J}$ of $\operatorname{int}(M)$ with compact support $S \subseteq \operatorname{int}(M)$, the **isotopy extension theorems** assert the existence of a diffeotopy $J$ of $M$ such that $J_t|_{\partial M} = \partial J_t$ and $J_t|_S = \tilde{J}_t|_S$ for each $t \in [0,1]$ (Hirsch, 1994, Thm 8.1.3, 8.1.4).

Let $M$ be a $k$-dimensional smooth manifold with boundary that is a subset of an $n$-dimensional smooth manifold with boundary $N$, such that the topology on $M$ is the subspace topology inherited from $N$. If the inclusion map $M \hookrightarrow N$ is a smooth embedding, then $M$ is called a **smoothly embedded submanifold with boundary** of $N$ (Lee, 2013, p. 120). When $\partial N = \varnothing$, such an $M$ has the property that each $x \in M$ is contained in a chart $(U,\varphi)$ for $N$ such that $\varphi(U \cap M)$ is an open subset of the intersection of a $k$-dimensional affine subspace with $\mathbb{H}^n$ (Lee, 2013, Thm 5.51). Conversely, if $\partial N = \varnothing$ and $M \subseteq N$ is any subset of $N$ with this property, then with the subspace topology, $M$ has a smooth structure making it into a $k$-dimensional smoothly embedded submanifold with boundary of $N$ (Lee, 2013, Thm 5.51).

If $M$ is a smoothly embedded submanifold with boundary of a smooth manifold with boundary $N$, then any Riemannian metric on $N$ canonically induces a Riemannian metric on $N$ (Lee, 2013, p. 333). Thus, if $N = \mathbb{R}^n$, a smoothly embedded submanifold with boundary $M \subseteq N$ canonically inherits a Riemannian metric from the Euclidean inner product, since the latter is a Riemannian metric on $\mathbb{R}^n$ called the **Euclidean metric** (Lee, 2013, Ex. 13.1). In this case, we refer to the finite Borel measure $\mu$ on $M$ determined from the Euclidean-induced metric as the **intrinsic measure**. If such an $M$ is $k$-dimensional, then $\mu(S)$ is simply the $k$-dimensional volume of $S$. In particular, $\mu(S)$ is the length of $S$ when $k = 1$, the surface area of $S$ when $k = 2$, the volume of $S$ when $k = 3$, and so on.

Given Euclidean spaces $\mathbb{R}^\ell$ and $\mathbb{R}^m$, the **compact-open topology** on the space $\mathcal{C}(\mathbb{R}^m, \mathbb{R}^\ell)$ of continuous maps $\mathbb{R}^\ell \to \mathbb{R}^m$ is defined as follows. A subset $\mathcal{S} \subseteq \mathcal{C}(\mathbb{R}^m, \mathbb{R}^\ell)$ is **open** if, for each $f \in \mathcal{S}$, there is a compact set $K \subseteq \mathbb{R}^\ell$ and $\varepsilon > 0$ such that any $g \in \mathcal{C}(\mathbb{R}^m, \mathbb{R}^\ell)$ satisfying $\max_{x \in K} \|f(x) - g(x)\| < \varepsilon$ belongs to $\mathcal{S}$ (Hirsch, 1994, p. 58). The **composition map**

$$\mathcal{C}(\mathbb{R}^n, \mathbb{R}^m) \times \mathcal{C}(\mathbb{R}^m, \mathbb{R}^\ell) \to \mathcal{C}(\mathbb{R}^n, \mathbb{R}^\ell), \qquad (g, f) \mapsto g \circ f$$

is continuous with respect to the compact-open topologies (and the product topology on the domain) (Hirsch, 1994, p. 64, Ex. 10(a)).

### B.4 Algebraic topology

The **standard $n$-simplex** $\Delta^n \subseteq \mathbb{R}^{n+1}$ is the convex hull of the standard basis vectors for $\mathbb{R}^{n+1}$, equipped with the subspace topology (Hatcher, 2002, p. 103).

Let $X$ be a topological space. A **singular $n$-simplex** in $X$ is a continuous map $\sigma \colon \Delta^n \to X$ (Hatcher, 2002, p. 108). A **singular $n$-chain** with coefficients in the abelian group $\mathbb{Z}_2 \coloneqq \mathbb{Z}/2\mathbb{Z}$ is a finite formal linear combination $\sum_i n_i \sigma_i$, where each $n_i \in \mathbb{Z}_2$ and each $\sigma_i$ is a singular $n$-simplex in $X$ (Hatcher, 2002, pp. 153, 108). The set of all singular $n$-chains in $X$ is an abelian group $C_n(X; \mathbb{Z}_2)$ (Hatcher, 2002, p. 153). There are well-defined group homomorphisms $\partial_n \colon C_n(X; \mathbb{Z}_2) \to C_{n-1}(X; \mathbb{Z}_2)$, called **boundary operators**, that satisfy $\partial_n \circ \partial_{n+1} = 0$ (Hatcher, 2002, pp. 153, 108). Thus, the image $B_n(X; \mathbb{Z}_2)$ of $\partial_{n+1}$ is contained in the kernel $Z_n(X; \mathbb{Z}_2)$ of $\partial_n$, so the $n$-**th singular homology group with coefficients in** $\mathbb{Z}_2$ is well-defined as the quotient group (Hatcher, 2002, pp. 153, 108)

$$H_n(X; \mathbb{Z}_2) \coloneqq Z_n(X; \mathbb{Z}_2)/B_n(X; \mathbb{Z}_2).$$

From a certain point of view, $H_n(X; \mathbb{Z}_2)$ counts the number of "$n$-dimensional holes" in $X$ (cf. Hatcher (2002, p. 100, Thm 2.27, p. 153)).

Let $X$ be a $k$-dimensional manifold (i.e., without boundary). It is a fact that $H_n(X; \mathbb{Z}_2) = 0$ for all $n \geq k$ when $X$ is noncompact, and that $H_n(X; \mathbb{Z}_2) = 0$ for all $n > k$ when $X$ is compact (Hatcher, 2002, p. 236, Prop. 3.29). Unlike the noncompact case, $H_k(X; \mathbb{Z}_2) = \mathbb{Z}_2 \neq 0$ when $X$ is compact (Hatcher, 2002, p. 236).

Any continuous map $f \colon X \to Y$ between topological spaces induces a well-defined homomorphism $f_* \colon H_n(X; \mathbb{Z}_2) \to H_n(Y; \mathbb{Z}_2)$ for each integer $n$ by sending the equivalence class of an $n$-chain $\sum_i n_i \sigma_i$ to the equivalence class of the $n$-chain $\sum_i n_i f \circ \sigma_i$ (Hatcher, 2002, p. 111).

A **homotopy** is a continuous map $h \colon [0, 1] \times X \to Y$, and is a **homotopy from** $f \colon X \to Y$ **to** $g \colon X \to Y$ if $f = h(0, \cdot)$ and $g = h(1, \cdot)$ (Hatcher, 2002, p. 3). Two maps $f, g \colon X \to Y$ are **homotopic** if there is a homotopy from $f$ to $g$ (Hatcher, 2002, p. 3).

A fundamental result called **homotopy invariance** asserts that homotopic maps $f, g \colon X \to Y$ induce the same homomorphisms on homology, i.e., $f_* \colon H_n(X; \mathbb{Z}_2) \to H_n(Y; \mathbb{Z}_2)$ coincides with $g_* \colon H_n(X; \mathbb{Z}_2) \to H_n(Y; \mathbb{Z}_2)$ for each integer $n$ (Hatcher, 2002, Thm 2.10, p. 153).

