# OpenReview forum: "Why should autoencoders work?"
_TMLR — Accepted by TMLR_

### Review · Reviewer_LRt6 · 2023-11-17

**Summary Of Contributions:**

The authors offer a theoretical justification to the observed success of deep NN auto-encoders (AEs) despite known topological obstructions in regards to the underlying (or assumed) data manifold. In summary, the authors appeal to known results in differential geometry substantiating the existence of continuous maps, e.g., an encoder (F) and a decoder (G), that yield the desired AE for all but a subset of *small measure*. This subset of small measure, where the AE does not preserve the data manifold, allow flexibility against the topological obstructions mentioned earlier. The derivations are supported by a simple numerical example of two interlocked 1D circles in 3D, where the learned AE effectively ends up cutting both circles, destroying the input topology.

**Audience:**

Yes

**Claims And Evidence:**

Yes

**Requested Changes:**

**Technical issues:**
- Please (**T1**) add a conclusions sections where lingering theoretical and practical issues are clearly outlined.
  - For one thing, many approaches use the latent embeddings to *walk* along the *learned manifold*, e.g., generating/interpolating new samples. It would be interesting to (**T2**) discuss the implication of the implicit cutting performed by the AE as in the example shown.
  - I can offer some more suggestions in a follow up comment.
- Section 3:
  - Please (**T3**) indicate that the ADAM optimizer was used. It would help to further highlight this choice, as well as any other insights/challenges from trying different architecture as alluded to in the text, in the future work section.
    - **most neural net training algorithms, including the one that we employ (TensorFlow) are stochastic** -- I would not call TF a training algorithm per se; please revise.
- Surfacing some presentation issues that are potentially more serious.
  - Page 2:
    - **It** *asserts that [..] with error smaller than $\varepsilon$ on S*
      - Please (**T4**) clarify that **it** refers to the PAC theorems cited, and not Theorem 1.
    - *A related idea seems to have appeared in (Hecht-Nielsen, 1995, Fig. 4)*
      - Please (**T5**) briefly describe the idea and how it's related to the discussion.
    - Around Lemma 2
      - Please (**T6**) double-check where the requirement of a finite-union is needed. If there's no need to speak of *any union* or *countable* union, then it's better to stick to one form. Specifically, Theorem 1 is stated for finitely-many submanifolds. On the other hand, the proof of Lemma 2 starts by countability as an implication. It would help to simplify this so long as it does not impact the main results.
      - Please (**T7**) define the reach before Lemma 2, as later used Section 4. A brief definition and forward reference could be enough.
      - Please (**T8**) add an adequate description of the outer regularity criterion, before engaging in the proof of Lemma 2.
  - Proof of Theorem 1
    - Please (**T9**) state clearly the properties assumed and implied on the first line of the proof. It is unnecessarily confusing to refer to sentences on different pages.
  - Theorem 3: seeing that the proof is by contradiction, it would help to still (**T10**) provide intuition as to why the result is optimal.

**Presentation issues:**
- I'm pointing out locations where a typical ML reader may have a hard time following (i.e., speaking on behalf of your readers):
- To avoid repetition, please note:
  - IMHO, adding a reference to an entire chapter (or an exercise) in an advanced math textbook may serve as evidence enough for an expert, but is not helpful to the non-expert or beginner.
  - See also **Tell them what you'll tell them** from [Ten Simple Rules for Mathematical Writing](https://www.mit.edu/~dimitrib/Ten_Rules.html)
- Page 1:
  - *Clearly such F, G exist if and only if K is homeomorphic to a k-dimensional submanifold*
    - Please (**P1**) add 1-2 sentences gently defining homeomorphism, explaining the iff intuitively.
- Page 2:
  - *Recall that there is an intrinsic notion of “measure zero” subsets of a smooth manifold with boundary*
    - Please (**P2**) add 1-2 sentences gently defining the intrinsic measure and measure zero subsets.
  - Please (**P3**) add 1-2 sentences gently defining the cut locus and a diffeomorphism (explaining the $H: M \setminus C_0 \approx \mathbb{R}^k$ notation), before engaging in the proof of Lemma 1.
  - Proof of Lemma 1
    - Please (**P4**) state in plain language what the guarantee $J(S) \cap C_0 = \emptyset$ means.
    - Please (**P5**) state in plain language what the solution $C = J^{-1}(C_0)$ means.
      - It would help to inform the reader of the intended plan of defining the desired $C$ by mapping a cut locus $C_0$.
    - Please (**P6**) add 1-2 sentences gently defining the double $DM$.
    - Please (**P7**) add 1-2 sentences gently defining what it means to be a diffeomorphism *onto* a given set.
- Page 3:
  - Please (**P8**) add an adequate description of Sard's theorem before engaging in the proof of Lemma 2.
  - Please (**P9**) add 1-2 sentences gently defining the smooth extensions used in the proof of Lemma 2.
- Proof of Theorem 1:
  - Please (**P10**) add 1-2 sentences gently defining density in the compact-open topology, and inform the user of its role in the proof, before engaging in the proof.
- Section 4:
  - Please (**P11**) add 1-2 sentences gently defining smallness in the Baire sense (meagre), and how that enables the conclusion on the following sentence *by continuity*.
  - Please (**P12**) add an adequate definition of Cech cohomology, how the choice of the Abelian group matters, why we're only interested in the $k$-th cohomology group, what it means when it's not $0$, and why this is all showing up here in this discussion of manifolds and diffeomorphisms (see next point).
  - Please (**P13**) add 1-2 sentences gently defining homotopy and homotopy invariance, before engaging in the proof of Theorem 3.

**Strengths And Weaknesses:**

**Strengths:**
- Concise
- Supported by an intuitive and effective example

**Weaknesses:**
- The theoretical insights rest of technical results that aren't made digestible for the broad audience.
  - The authors may be targeting a different audience, which is fine. Though it's certainly a loss to miss out on teaching something to the greater ML community.
  - Just a couple references to where I'm coming from:
    - [On Proof and Progress in Mathematics](https://www.ams.org/journals/bull/1994-30-02/S0273-0979-1994-00502-6/?active=current),
    - [Why Mathematical Proof Is a Social Compact](https://www.quantamagazine.org/why-mathematical-proof-is-a-social-compact-20230831)
- Indeed, the proofs lean on being a bit too terse.
- While conciseness is appreciated, it leaves a lot unsaid. An extra section with conclusions and open questions would be great.

---

> ### Comment · Reviewer_LRt6 · 2024-01-31
> **Comments on the revised version**
>
> - Section 1
>   - Suggest a separate remark for "One should emphasize that Theorem 1 [..] does not imply numerical learning algorithms will always succeed"
>   - Last line should probably appear immediately after Theorem 1.
> - Section 2
>   - Recommend to begin the section by a paragraph outlining the "game plan" for proving Theorem 1. Then, present each lemma using a short paragraph with a basic account of why it's needed and possibly a sketch of the proof or its underlying intuition.
>   - Lemma 1:
>     - It would help to break down the (presentation of the) proof into steps: (1) setup, (2) construction of C, (3) C is closed and connected, (4) C has measure zero, (5) C admits a smooth embedding.
>     - Please add references to the appendix B3 for the definition of hyperbolic equilibrium points, and the fact that the basin of attraction is open and connected.
>     - I take it "initial conditions" simply mean "points" on the manifold along any given flow.
>   - Lemma 2:
>     - It would be nice to assure the reader that the proof still makes sense if $\partial M$ is empty, i.e., only less constraints on $J_t$ which the proof only defines through extension.
>     - Would also help to mention that $t \in [0, 1]$ to elucidate $J_t$ becoming $J_1$.
>   - Lemma 3:
>     - The symbol $n$ was taken for the ambient dimension.
>     - Is it the finiteness of $S$ or the finiteness of the number of components of $K$, that is needed for the existence of $N$?
>     - Compactness of $C$ wasn't explicitly established earlier.
>   - Theorem 1:
>     - Since $K$ is compact, (and by the) density of $\mathcal{F}$ [..] (there exists) $F \in \mathcal{F}$
> - Section 4:
>   - I would prefer to have Remark 8 appear at the beginning before getting into any derivations.
> - Section 5:
>   - reconstruction uniform error -> uniform reconstruction error
>   - In discussing time series data, I expected citations of works on reduced order modeling or perhaps dynamic mode decomposition.
> - Appendix:
>   - First paragraph of Appendix B is missing $B.4.

---

> ### Comment · Reviewer_LRt6 · 2024-01-31
> **Walking along learned manifolds**
>
> Some examples:
> - Shu, Zhixin, Mihir Sahasrabudhe, Riza Alp Guler, Dimitris Samaras, Nikos Paragios, and Iasonas Kokkinos. "Deforming autoencoders: Unsupervised disentangling of shape and appearance." In Proceedings of the European conference on computer vision (ECCV), pp. 650-665. 2018.
> - Qian, Kaizhi, Yang Zhang, Shiyu Chang, Xuesong Yang, and Mark Hasegawa-Johnson. "Autovc: Zero-shot voice style transfer with only autoencoder loss." In International Conference on Machine Learning, pp. 5210-5219. PMLR, 2019.
> - Tretschk, Edgar, Ayush Tewari, Michael Zollhöfer, Vladislav Golyanik, and Christian Theobalt. "Demea: Deep mesh autoencoders for non-rigidly deforming objects." In Computer Vision–ECCV 2020: 16th European Conference, Glasgow, UK, August 23–28, 2020, Proceedings, Part IV 16, pp. 601-617. Springer International Publishing, 2020.

---

> > ### Author Response · Authors · 2024-02-03
> > **"Walking on mainfolds"**
> >
> > The referee asked if we could comment on how our work might apply to "walking along learned manifolds". If we understand the referee's question, this fits within the general area of using AE's for "interpolation". To summarize (and perhaps oversimplify) that field, the basic idea is as follows. Given two manifold points $p$ and $q$, one first performs "interpolation" in the latent space between $F(p)$ and $F(q)$, meaning that one draws a path between these two points (say, a straight line, if the segment between $F(p)$ and $F(q)$  is included in the image). Through decoding, one obtains a path between $G(F(p))$ and $G(F(q))$ (that is, between approximately $p$ and approximately $q$) in the original space. The hope is that this path will remain (approximately) in the manifold $K$, by construction.  In an image processing context, this procedure allows one to create a smooth transition between two images while preserving general characteristics (represented by the curve staying in, or near, a manifold K), for example, if $p$ and $q$ represent faces, one would have a set of possible faces that lie "between" the two given ones. However, the topological obstruction quantified by the reach tells us that some points (images, in the example) along the path will fall very far from $K$, and thus will be "unrealistic". It would be an interesting direction of research to think of ways to impose additional constraints on the small-measure sets $K_0$ that need to be "cut out" so that, say, for a finite set $S$, for each two points $p$ and $q$ in $S$ there is some path that does not intersect $K_0$. We prefer not to say anything in the paper, since we do not have precise statements, much less results, to state.

---

> > > ### Comment · Reviewer_LRt6 · 2024-02-03
> > > **Request #T2**
> > >
> > > It is ok to disagree on this point. Below I explain my take and a few more comments.
> > >
> > > Explaining why AEs "work" against known topological obstructions is already a valuable contribution, with clear practical relevance. I'm guessing, having read the manuscript and comments, that the authors prefer to keep their paper within this subset of literature on topological obstructions and closely related areas.
> > >
> > > However, "work" can mean different things in different circumstances. A lot of my requests aimed at making the work more accessible with the overarching goal of making it more relevant to a broader reader base.
> > >
> > > Pointing out where AEs are unable to overcome those known topological obstructions is important to highlight. It is not immediately obvious to all readers, e.g., those who will still be inspired by those powerful examples utilizing AEs for interpolation over all sorts of modalities.
> > >
> > > I believe even a brief mention of those issues only adds to the value of this paper, and could lead to more useful developments as the authors shared; see also first comment below.
> > >
> > > A few more comments:
> > > - I wonder if it's possible to combine multiple AEs, with mild independence criteria such as random initialization, to effectively produce mappings capable of straddling those necessary cuts. That is, recovering the smoothness of the walk by continuation in alternate latent spaces, or a product manifold. How many such AEs would be needed? O(1)? O(N)?
> > > - While I appreciate the new discussion and future work section, I feel it's too broad like what one would read in a thesis rather than an article. It would recommend to distill this section into, say, 3 open questions or directions that are most relevant. Perhaps the remaining questions/directions can be briefly highlighted in one last paragraph.

---

### Review · Reviewer_rGgR · 2023-11-22

**Summary Of Contributions:**

This paper proves a universal approximation theorem of auto-encoder  for approximating continuous maps between general manifolds with possibly different dimensions.

**Audience:**

No

**Broader Impact Concerns:**

oNone.

**Claims And Evidence:**

Yes

**Requested Changes:**

As mentioned above, the novelty of the paper is low. I suggest adding a proof of UAP property for neural network-based auto-encoders in the new version.

**Strengths And Weaknesses:**

Strength: The approximation property of auto-encoder underpinnings many applications in machine learning. Therefore studying the UAP property is certainly of great importance.  The main result shows that auto-encoder defined by the composition of two continuous functions with the latent dimension $k$ less than the input dimension $n$ can approximate the identity map on $R^n$ on the whole manifold expect an arbitrary small set.

Weakness: My major concern is the novelty of the paper. The proofs of the main results are merely a convex combination of some facts from differential geometry and topology. Also, practical auto-encoders make use of deep neural networks in encoders and decoders. The current paper only discussed the general composition of continuous maps, rather than the neural auto-encoders, which make the paper less interesting to the ML community. The paper would have been significantly improved if a proof of UAP property for neural network-based auto-encoders is included.

---

### Review · Reviewer_XucU · 2023-12-20

**Summary Of Contributions:**

Autoencoder is an important class of neural networks in deep learning, which aims to capture data with $G(F(x)) = x$ with encoder $F: \mathbb{R}^n \to \mathbb{R}^k$ and decoder $G: \mathbb{R}^k \to \mathbb{R}^n$. Since $F$ and $G$ are always continuous, there could be topological obstructions making this goal impossible: the support of data may not be homeomorphic to a k-dimensional submanifold in general. Still, autoencoders work well in practice.

This paper provides an approximation theorem to explain this apparent paradox. In particular, they show that autoencoders are actually capable of approximately representing data even if the data is supported on a union of compact submanifolds having dimension less than or equal to $k$.

**Audience:**

Yes

**Claims And Evidence:**

Yes

**Requested Changes:**

* The paper title "Why do autoencoders work?" seems to suggest that the paper would provide an extensive study on why autoencoders work, but the paper's content is limited to the expressive power of autoencoders and does not provide much insight into the training dynamics. Also no experiments are conducted in a real-world setting. I would suggest changing the title to more accurately reflect the paper's essence. For example, "How do autoencoders represent discontinuously distributed data?" could be a good title.
* It would be nice if the authors could add a conclusion section to summarize the main results and propose some future directions.

**Strengths And Weaknesses:**

Strengths:
* The main theorem is very solid. Although it could be intuitive that autoencoders may connect disjoint submanifolds together by creating break points and adding something among them, it is not a very easy task to make it completely rigorous in math. This paper uses several tools from differential geometry to establish their claim.
* A simple example/experiment is provided in Section 3 to illustrate the idea.

Weaknesses:
* For readers with limited knowledge about differential geometry, the proof may be a bit hard to read. If the authors are willing to improve the readability, I would suggest adding a preliminary section to elaborate on the concepts/theorems to be used in the proof, e.g., cut locus, the identification used in Lemma 1, intrinsic measure, Sard's theorem, the extension theorems used in Lemma 2, compact-open topology, etc.
* As a common limitation of all other papers studying the expressiveness of neural networks, it remains a challenging task to prove that neural networks can indeed fully utilize their expressive power and learn the constructed function via gradient-based training methods.

Typo:
* Lemma 2: In Line 4 of Paragraph 2, is F a smooth extension of F_0 or F itself?

---

### Decision · Action_Editor_mQh8 · 2024-02-12

**Recommendation:** Accept as is

**Comment:**

This paper studies the expressive power of autoencoders, highlighting topological obstructions to perfect reconstruction, and then proving with techniques from differential topology that these obstructions can be made to have a negligibly small effect. The reviewers found the results to be rigorous, and, after revision, the proofs to be reasonably approachable for a machine learning audience.

The reviewers' main concerns relate to the practical utility and significance of the results, as the analysis does not address the question of whether the autoencoding functions can actually be learned. I agree with the authors that answering this question is out of scope but would be an interesting future direction. On the other hand, some aspects of the paper's messaging (primarily the title) seem to suggest that this paper might in fact address practical aspects of autoencoder training, including this question. As such, I would encourage the authors to consider a slightly more conservative title, though ultimately the decision is theirs.

**Audience:**

Yes, some members of the community will be interested in this paper.

**Claims And Evidence:**

Yes, the claims in the submission are well supported.